

# ICON-ART 2.1 - A flexible tracer framework and its application for composition studies in numerical weather forecasting and climate simulations

Jennifer Schröter[1], Daniel Rieger[1,3], Christian Stassen[1,4], Heike Vogel[1], Michael Weimer[2],
Sven Werchner[1], Jochen Förstner[3], Florian Prill[3], Daniel Reinert[3], Günther Zängl[3], Marco Giorgetta[5],
Roland Ruhnke[1], Bernhard Vogel[1], and Peter Braesicke[1]

[1]Institute of Meteorology and Climate Research, Karlsruhe Institute of Technology, Karlsruhe, Germany
[2]Steinbuch Centre for Computing, Karlsruhe Institute of Technology, Karlsruhe, Germany
[3]Deutscher Wetterdienst, Offenbach, Germany
[4]now at: ARC Centre of Excellence for Climate System Science, School of Earth Atmosphere and Environment, Monash University, Australia
[5]Max Planck Institute for Meteorology, Hamburg, Germany

*Correspondence to:* Jennifer Schröter (jennifer.schroeter@kit.edu)

**Abstract.**

Atmospheric composition studies on weather and climate time scales require flexible, scalable models. The ICOsahedral Nonhydrostatic model with Aerosols and Reactive Trace gases (ICON-ART) provides such an environment. Here, we introduce the most up-to-date version of the flexible tracer framework for ICON-ART and explain its application in one numerical weather

forecast and one climate related case study. We demonstrate the implementation of idealised tracers and chemistry tendencies of different complexity using the ART infrastructure. Using different ICON physics configurations for weather and climate with ART, we perform integrations on different time scales, illustrating the model's performance. First, we present a hindcast experiment for the 2002 ozone hole split with two different ozone chemistry schemes using the numerical weather prediction physics configuration. We compare the hindcast with observations and discuss the confinement of the split-vortex using an

idealised tracer diagnostic. Secondly, we study AMIP type integrations using a simplified chemistry scheme in conjunction with the climate physics configuration. We use two different simulations: The interactive simulation, where modelled ozone is coupled back to the radiation scheme and the non-interactive simulation that uses a default background climatology of ozone. Additionally, we introduce a chemical source term for water vapour for the interactive simulation. We discuss the impact of stratospheric ozone and water vapour variations in the interactive and non-interactive integrations on the water vapour tape

recorder, as a measure of tropical upwelling changes. Additionally we explain the seasonal evolution and latitudinal distribution of the age of air. The age of air is measure of the strength of the meridional overturning circulation with young air in the tropical upwelling region and older air in polar winter downwelling regions. We conclude that our flexible tracer framework allows for tailor-made configurations of ICON-ART in weather and climate applications that are easy to configure and run well.





## 1   Introduction

Next generation global modelling systems allow simulations with high horizontal resolutions, up to grid spacings of a few hundreds of meters. Thus, the vertical momentum equation is fully retained and allows for non-hydrostatic motions. In general, unified modelling systems for weather and climate applications have several advantages. They use the same physical parameterisations and the same dynamical core across a wide range of temporal and spatial scales. Thus, simulations ranging from a few hours to hundreds of years are very consistent and advances of one scale or application can be more easily transferred to another.

The extended modelling system ICON-ART (ICOsahedral Nonhydrostatic - Aerosols and Reactive Trace Gases;  Rieger et al., 2015) is an example of a next generation system. ICON (Zängl et al., 2015) is a joint development of Deutscher Wetterdienst (DWD) and Max Planck Institute for Meteorology (MPI-M). The dynamical core of ICON is based on the non-hydrostatic formulation of the vertical momentum equation. Thus, ICON allows simulations with high horizontal resolutions, up to grid spacing of a few hundreds of meters. However, while the dynamical core of ICON is unified individual applications like large eddy simulations, numerical weather prediction or climate projections use currently different parameterisations of physical processes (Dipankar et al., 2015; Heinze et al., 2017).

ICON offers the possibility of local grid refinements, also called 'nesting'. The nesting provides the option of two-way interactions between the (global) coarse domain and the higher resolution local domain. Hierarchical nesting and several domains at the same nesting level are possible. ICON already shows benefits in various application fields, e.g. in the HD(CP)$^2$ project. Selected results of the HD(CP)$^2$ project are described in Heinze et al. (2017). Within this project the configuration of the LES physical configuration has been developed and is used to achieve a better understanding of, for example, processes that are related to precipitation. Results will help to improve future climate projections on coarser resolutions.

Since January 2015 ICON runs operationally at DWD for weather forecasts on the global scale with a grid spacing of $13 \, \text{km}$. Since June 2015, the local grid refinement over Europe with a horizontal resolution of $7 \, \text{km}$ is in operational use as well.

ART is an additional module for ICON, developed at the Karlsruhe Institute of Technology. It contributes to the goal of a unified global next generation modelling system with a variety of applications in the field of atmospheric composition sciences. ICON-ART extends the numerical weather and climate prediction system ICON with (gas phase) chemistry, aerosol dynamics and related feedback processes. A comprehensive treatment of feedback processes between chemistry, aerosols, clouds and radiation will be included in many recent and future studies (Gasch et al., 2017; Rieger et al., 2017; Weimer et al., 2017; Eckstein et al., 2017).

Here, we will discuss the unified tracer framework of ICON-ART that allows to define and specify tracer initialisations and a coupling of individual chemical mechanisms and specific process modules. Depending on the requirements of the application field, like large eddy simulations, numerical weather predictions and climate simulations ICON-ART can run with the different existing physics configurations.

Given the multitude of challenges we are facing, models have to be capable of being readily changed. Therefore, it is of high importance to provide a tracer framework that is flexible and suitable for a large variety of different applications. For the





development of next-generation modelling systems, the requirements for modern high performance computer architectures also have to be considered. Nowadays, numerical models are integrated on massively parallel architectures with up to $10^6$ cores. Unstructured grids like the icosahedral ICON grid show advantages regarding the performance on current high-performance computing systems (e.g. Zängl et al., 2015). Taking into account the increase of computational power over the last years, it is

obvious that more diagnostic and prognostic tracers will be included in future simulations than before. With ICON-ART 2.1 we introduce a new flexible tracer framework, which meets the demands of a next generation modelling system. There is full flexibility in defining the tracers and associated characteristics. The set of tracers can be tailored for individual experimental setups of different complexity. The model allows for changes in the set of tracers without any recompilation. The ability to replace the common usage of namelist structures, previously used by ICON-ART and other models (e.g. Morgenstern et al.,

2017; Baklanov et al., 2014) is fully supported.

Here, we describe the concept of the new ICON-ART tracer framework. The technical description is followed by examples of definitions for chemical and passive tracers in ICON-ART in Section 5. In Section 5 we give an overview of different applications of ICON-ART using the flexible tracer framework. These applications include simulations with the numerical weather prediction physics configuration and simulations with the climate physics configuration. We finish with some concluding re-

marks and an outlook.

## 2   The ICON-ART tracer framework

Focussing on the simulation of the chemical composition of the Earth's atmosphere, it is highly desirable to design individual experiments on different temporal and spatial scales. For example, if an experiment mainly focusses on tropospheric chemical composition, it is useful to only define tropospheric chemical tracers. For long term simulations, it might be advantageous to

calculate a subset of chemical tracers leading to a less expensive calculation, in terms of computational time. Chemical tracers with a short life time, compared to dynamical time scales can be neglected in the transport processes.

With ICON-ART, we introduce a flexible tracer framework with a minimum of predefinitions. The user has the ability to define a unique set of tracers, tailored to the requirements of the model experiment planned. The new tracer framework builds on structures which are already implemented in the basic ICON framework and shared by different available physical

configurations. These configurations are the LES physics configuration (Dipankar et al., 2015; Heinze et al., 2017), the NWP physics configuration and the climate physics configuration based on ECHAM6 parametrisation (Stevens et al., 2013). The new tracer framework allows flexible coupling to the selected physical configuration. Thus, the composition coupling can be performed without subsequent changes. The ART code is not changed when the physics configuration is changed.

This allows an independent investigation of atmospheric processes using different physical configurations.

### 2.1   Technical description of the tracer framework

The technical framework of ICON-ART is based on Fortran 2003, which is essential for the new tracer framework. Classes can give the ability to overcome the stringent matching of data types. Items only have to match the data type or any extension of

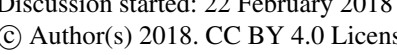


this type when it is declared with `CLASS(type)` where `type` is a derived data type (Chapman, 2008). If the `CLASS` matches more than one data type, it is called polymorphic. Within ICON-ART, we are using these polymorphic objects to extend the existing tracer structure by additional metadata. The second main feature that we are using are linked lists. Linked lists allow for new features like passing a reference to the next list element or adding and deleting elements.

Within the scope of this paper, we are working with two different types of tracers: passive and chemical active ones. An example for a passive tracer is a constituent, that does not have any impact on other tracers of the (thermo-)dynamical of the simulated system. Passive tracers have pre-defined sources and or only fixed initial conditions. They change by transport and do not interact with other tracers or processes. Chemical active tracers experience sources and sinks while being transported and can participate in feedback processes.

Chemical tracers and passive tracers are distinguished by their associated metadata. In our example, passive tracers have no chemical loss term to be considered. In other words, the lifetime of passive tracers is infinite. Information about the lifetime is used for simple chemical tracers that experience an idealised loss while being transported. Of course, the information dealing with transport should be provided for both tracer types. A flexible aerosol dynamics module making use of the flexible tracer framework is currently under development.

## 2.2 Storage of metadata information

We extend the existing ICON-ART functionality by the usage of a key-value storage. The storage is based on string comparisons using generic hash tables. For every tracer, a unique key of fixed size is defined. This represents the search key. The table is constructed like a dictionary. Tracer information can be looked up fast, using this search key. Hash tables provide foundation for a generic and flexible concept to read and store metadata. At model initialisation one key-value storage for each tracer is

initialised by

```
CALL storage%init(lcase_sensitivity=.FALSE.)
```

In this case, the dictionary entries are not case sensitive. The storage unit has two fundamental subroutines, these are

`storage%put` and `storage%get`. The first creates a new dictionary entry, and the second searches for a key and gives the connected entry. In the next step, this storage unit has to be filled. Accepted types of storage values are reals, integers or characters.

### 2.2.1 Reading in metadata information from XML files

At one point in time, all numerical models need to find an answer to the question how to transfer text based information

(e.g. tracer metadata) into the model's program code. The text based format XML - eXtensible Markup Language (see W3C-Recommendation, 2017), gives the developer and the user the ability to store and transport information in a structured way.

XML has only few mandatory rules, e.g. there has to be exactly one root-element. The framework in which the information is structured can be freely chosen. The structure itself allows for a human readable format. ICON-ART uses the Fortran interface





TIXI (https://software.dlr.de/p/tixi/home/) to read the XML file. The TIXI interface includes a flexible mechanism for XML file read in. Since the scripting language of XPath is used, the navigation through an XML document is an easy task to perform. The XML reading routine can be structured in the same way like someone would read a document and would remember the content in a most natural way. An example structure of such an XML input file for the tracer structure is the following:

```
<?xml version="1.0" encoding="UTF-8"?>
<!DOCTYPE tracers SYSTEM "tracers_gp.dtd">

<tracers>

      <chemical id="TRO3">

      <mol_weight type="real"> 4.800E-2 </mol_weight>

      <lifetime type="real"> 2592000 </lifetime>

      <transport type="char"> stdchem </transport>

<init_mode type="int"> 0 </init_mode>

      <unit type="char"> mol mol-1 </unit>

      <long_name type="char"> ozone </long_name>

      </chemical>

    </tracers>
```

The XML file is scanned automatically. For the realisation of this feature, it is necessary to predefine the type of input. Every tag has a mandatory type definition, being `char, int` and `real`. The first word in brackets, `<tracers>` is called XML-tag. The tag `<chemical id="TRO3">` has the additional attribute `id` for the tag `chemical`. To identify a specific tracer, the system uses the given attribute (e.g. `id`). Tags are used to build up the metadata structure. It is a key-value storage where

the tag (e.g. `mol_weight`) is the key with the value, e.g. `4.800E-2` for ozone. The number `4.800E-2` is then stored in the metadata structure of ICON-ART. At this point it should be noted that there are two kinds of metadata: necessary and optional metadata. Necessary metadata depend on the polymorphic type. Passive tracers do have different necessary metadata than chemical tracers. The optional metadata are read in automatically. Every tag in the XML file is translated into an entry in the key-value storage.

The structure of the ART tracer framework is shared with the core ICON tracer framework and only expanded in cases needed. In addition to that, tracers can share attributes, and are distinguished by additional attributes. For example, ozone appears in two different subroutines for chemistry. The first, using a lifetime based mechanism, and the second, using a full gasphase approach.

The tag `<mol_weight>` can be used for unit conversion within a subroutine and is given in $\mathrm{kg\,mol^{-1}}$. The tag `<lifetime>`

is used for simplified integration methods for a given life time of a species, in seconds. Some chemical substances or passive tracers do not have to be transported thus the tag `<transport>` ensures that this information is transferred to the program





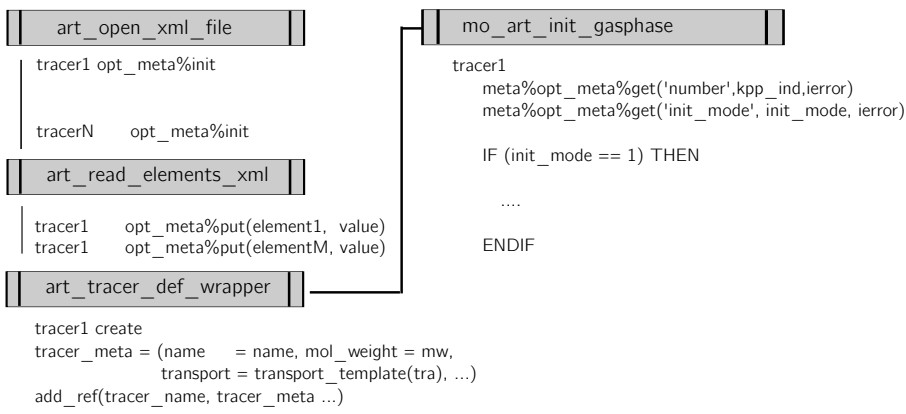

**Figure 1.** Schematic describing the tracer framework and the tracer definition in ICON-ART.

code and the transport is switched on or off, respectively. In addition to that, templates can be defined. In this case, a template named `stdchem` is chosen. At model start up, this template is translated into a specific selection of horizontal and vertical transport schemes. Each scheme stands for a specific numerical discretisation of the mass continuity equation in horizontal or vertical directions. Currently, there are three different transport templates available: `off`, `stdaero` and `stdchem`. These

templates avoid the necessity to add a tracer advection scheme and flux limiter for each single tracer in the namelist. Hence, the values of the ICON-ART namelist parameters `ihadv_tracer`, `ivadv_tracer`, `itype_hlimit` and `itype_vlimit` are overwritten by the template. Specific information concerning the advection schemes mentioned can be found in the official ICON documentation. Specifying `off` will deactivate advective transport for this tracer completely.

The transport template `stdchem` uses the same advection schemes as `stdaero`. However, the considerably faster positive

definite flux limiters are used. The conservation of linear correlations is traded for a faster computation of the advection.

If chemical species need to be initialised, this is achieved via `<init_mode>` and a respective number corresponding to the initialisation scheme. If the `init_mode` is set to 0, no external initialisation data is used. With integers different from zero, data from other models, like EMAC (Jöckel et al., 2005) or MOZART (Emmons et al., 2010), interpolated on the horizontal icosahedral grid are read in and used. The vertical interpolation is performed online. Thus, the initialisation dataset stays

unchanged, regardless of the choice of model levels for the simulation.

The structure of the key-value storage for every tracer is built up automatically. This feature allows for an extensive flexibility which has to be controlled. In our case, attributes are used in a fixed manner, they define our basic framework of the tracer structure. The full flexibility, direct access, control mechanisms and data type distinction, are only a few advantages of this XML based procedure against other currently used strategies.

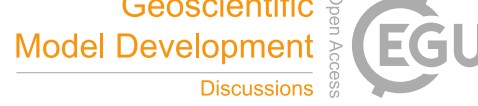

## 2.3 Construction of the tracer metadata structure

Within ICON-ART, the metadata, e.g. regarding transport or chemical properties, are read in using the XML interface. After-wards, the information is stored in the ICON-ART tracer framework. The individual steps of the tracer transport simulation in ICON and ICON-ART are described in Rieger et al. (2015). Figure 1 shows the calling structure of the tracer framework in

5   ICON-ART with regard to the tracer and metadata construction. Tracers have to be accessible for the prognostic and diagnostic state of ICON-ART. All tracers within the ICON-ART tracer framework share a common structure like the following:

```
TYPE t_tracer_meta
...
END TYPE t_tracer_meta
```

In case of chemical tracers, we want to provide additional information on, e.g. mol weight. The extension of the derived type looks like this:

```
TYPE, EXTENDS(t_tracer_meta) :: t_chem_meta
         REAL(wp) mol_weight
END TYPE t_chem_meta
```

In this case, the `mol_weight` is a mandatory information. Optional metadata is stored in as part of the basic structure `t_tracer_meta`. This container includes a key-value storage. This solution ensures, that the metadata container stays as

20   flexible as possible. The respective XML file is read in, and all information is processed and stored in the `opt_meta` container, see Figure 1. In the example of `mol_weight`, this attribute becomes a real variable accessible in the code. It can be accessed by using e.g.:

```
 SELECT TYPE(meta => info_dyn%tracer)
  CLASS IS (t_chem_meta)
   CALL meta%opt_meta%get('mol_weight', &
       & mol_weight, ierror)
  END SELECT
```

Here, `CALL meta%opt_meta%get(...)`

30   is the operation to access the key-value storage in the container.

The combination of linked lists, polymorphic classes and key-value storage allows the user to define tracers in a flexible way. Only those tracers given in an input file are read in, registered and appear in the modelling system. With this solution, the user is free to define any number of chemical or passive tracers which is only limited by the memory of the computing system.



## 2.4 Construction of XML tracer input file via MECCA

The XML file can either be edited manually by the user or be generated automatically by an external program. The tracer files used for the experiments described in Section 5, are shown in Appendix A. In this section, we demonstrate the XML file generation by an external program, which also extends the functionality of ICON-ART. In addition to the existing life time based chemistry approach (see Rieger et al., 2015) a full gas phase chemistry approach is added.

Construction of the full gas phase chemistry approach is done using the comprehensive and flexible atmospheric chemistry module MECCA (Module Efficiently Calculating the Chemistry of the Atmosphere; Sander et al., 2011). MECCA provides a set of chemical reactions covering the troposphere as well as the stratosphere. The base version of MECCA is extendible by own chemical reactions or an update of rate coefficients.

The MECCA preprocessing part has been extended by a routine constructing the XML file for the tracer module of ICON-ART. The numerical flexibility of MECCA is based on the KPP software (Kinetic PreProcessor; Sandu and Sander, 2006). KPP generates Fortran90 code which is used to solve the differential equations based on the given chemical reaction. This is the first step which is needed for KPP and thus for MECCA. In a second step, the numerical integrator is chosen. For our example, we use the Rosenbrock solver of the third order (Sandu et al., 1997). In the third step we use the driver part of

15 MECCA. The driver stands for the main program which calls the integrator, reads input data sets, and writes the results in the original MECCA model. Within ICON-ART we replaced this step by routines of ICON and ICON-ART. Only the integrator call element is maintained. Technical work has been done to ensure that a dictionary gets accessible for the translation between four dimensional chemical tracers in ICON-ART and one dimensional concentrations of chemical species in the KPP routines.

Additionally, other chemical reaction mechanisms provided by MECCA can be used within ICON-ART without changes

in the program code itself. It is sufficient, to read the new tracer XML file generated by the MECCA preprocessor. Finally, all standard reactions schemes provided by MECCA are accessible in ICON-ART, by only using MECCA as an external preprocessor. The model has to be recompiled once, but the user does not have to perform changes in the program code. Files that are changed by the preprocessor are copied to the respective ICON directory automatically. Photolysis rates are calculated using an updated version of CloudJ7.3 (Prather, 2015).

In the scope of this paper, we use a chemical reaction mechanism based on the extended Chapman cycle to demonstrate the functionality of the ICON-ART gasphase routine.

We defined the chemical reactions as shown in Table 1.

## 3 Integration of the ART module into ECHAM physics routines

The general coupling structure using NWP physics configuration is described in Rieger et al. (2015). The entry points of

30 the coupling to the climate physics configuration is different to the entry points for NWP. The general concept of clear code distinction using compiler directives, described in Rieger et al. (2015), remains the same.

Figure 2 shows an schematic overview of the calling strategy of subroutines with the ECHAM (climate) physics config­uration. Boxes marked partly orange are shared structures between ICON and ICON-ART. Boxes with an orange frame are





**Table 1.** Summary of the chemical reactions used for the extended Champman cylce simulation.

| Reactions used in the gas phase simulation - extended Chapman cycle |
| --- |
| $O_2 + O^1D + M \rightarrow O^3P + O_2 + M$ |
| $O_2 + O^3P \rightarrow O_3$ |
| $O_3 + O^3P \rightarrow 2O_2$ |
| $N_2 + O^1D \rightarrow O^3P + N_2$ |
| $OH + O_3 \rightarrow HO_2 + O_2$ |
| $HO_2 + O_3 \rightarrow OH + 2O_2$ |
| $HO_2 + OH \rightarrow H_2O + O_2$ |
| $H_2O + O^1D \rightarrow 2OH$ |
| $N_2O + O^1D \rightarrow O^3P + N_2O$ |
| $N_2O + O^1D \rightarrow 2NO$ |
| $NO + O_3 \rightarrow NO_2 + O_2$ |
| $NO_2 + O^3P \rightarrow NO + O_2$ |
| $NO_2 + O_3 \rightarrow NO_3 + O_2$ |
| $NO_3 + NO_2 \rightarrow N_2O_5$ |
| $NO_2 + OH + M \rightarrow HNO_3 + M$ |
| $HNO_3 + OH \rightarrow H_2O + NO_3$ |

| Photolysis reactions |
| --- |
| $O_2 + h\nu \rightarrow O^3P + O^3P$ |
| $O_3 + h\nu \rightarrow O^1D + O_2$ |
| $O_3 + h\nu \rightarrow O^3P + O_2$ |
| $NO_2 + h\nu \rightarrow O^3P + NO$ |
| $NO_3 + h\nu \rightarrow O^3P + NO_2$ |
| $N_2O_5 + h\nu \rightarrow NO_3 + NO_2$ |
| $HNO_3 + h\nu \rightarrow NO_2 + OH$ |





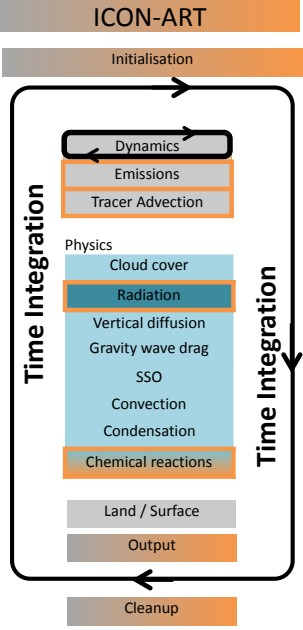

**Figure 2.** Schematic describing the process calling structure in the climate configuration of ICON-ART.

processes affected by ART tracer tendencies. Boxes with an orange background are ART routines. The model physics part is represented using different blue boxes. The radiation subprocess is depicted with a dark blue shaded box because the radiation process is called with a reduced frequency than other processes, in general. Chemical reactions are part of the physics routine in climate configuration but only active if ART is compiled and chemical routines are activated by a namelist parameter. All

processes marked with a blue shade can have an individual calling frequency. Most routines return a tendency and those tendencies are summed up at the end of the physics integration time step. The exceptions are the routine for condensation and total precipitation. Before both routines are called, the tendencies are updated.

## 4    Implementation of passive and chemical tracers in ICON-ART

It is computational expensive to couple global tropospheric and stratospheric chemical models with global meteorological mod-

els. Therefore, it is reasonable, especially at the beginning of the development of a model like ICON-ART, to use simplified parameterisations for the description of selected chemical species. Parameterisations of chemical species can save computational costs and can give a first overview of the general applicability of the model. In addition to parameterisations of chemical tracers, it is useful to define artificial passive tracers to investigate e.g. transport processes in the atmosphere. An example of a passive tracer is the vortex tracer, described at the end of the following section. Further, we describe a selection of parame-

terisations for the simulation of chemical tracers in the atmosphere. These tracers can also be used to investigate fundamental





composition-circulation feedbacks. The selection is an extension of the set of parameterisations described in Rieger et al. (2015) and Weimer et al. (2017).

### 4.1 Linearised ozone algorithm

Ozone is one of the most important chemical species regarding radiative heating in the middle atmosphere. Thus, ozone concentrations have to be simulated reasonably to address questions about transport processes determined by stratospheric processes (e.g. Braesicke and Pyle, 2003, 2004). ICON-ART, in its standard NWP configuration uses monthly climatological values derived from GEMS climatology (Global and regional Earth-system (Atmosphere) Monitoring using Satellite and in-situ data, (Hollingsworth et al., 2008)). The used ozone climatology is independent from the year of simulation. Here we move one step forward and use the ansatz for a simplified description based on McLinden et al. (2000). This parameterisation can be understood as a first order Taylor expansion of the stratospheric chemical rates. The ozone concentration tendency is linearised with respect to the local ozone mixing ratio, temperature and overhead ozone column density. The algorithm accounts for a more realistic vertical gradient than the provided climatology. For the troposphere, we assume a constant lifetime of 30 days.

The following differential equation describes the linearised approach:

$$
\begin{aligned}
\frac{\mathrm{d}\xi}{\mathrm{d}t} = {}& (P-L)^0 + \left.\frac{\partial(P-L)}{\partial\xi}\right|_0 (\xi - \xi^0) \\
& + \left.\frac{\partial(P-L)}{\partial T}\right|_0 (T - T^0) \\
& + \left.\frac{\partial(P-L)}{\partial c_{O_3}}\right|_0 (c - c_{O_3}^0) - \frac{1}{\tau_{psc}} \cdot \xi
\end{aligned}
\tag{1}
$$

Here, $\xi$ describes the ozone volume mixing ratio, $T$ the temperature in the respective grid box and $c_{O_3}$ the overhead ozone column. The term $(P-L)$ describes the ozone tendency, with $P$ the production term and $L$ the respective loss term. Climatological values are indicated with the superscript $^0$. The partial derivative with evaluation at the respective climatological value is marked with a subscript $_0$.

With this ansatz, heterogeneous processes are not taken into account. The regular linearised ozone (Linoz) parameterisation, does not include the last term $\frac{1}{\tau_{psc}} \cdot \xi$. To address the catalytic destruction by chlorine and bromine radicals in the presence of polar stratospheric clouds, the linearised ansatz is expanded by an additional loss term. Here, $\tau_{psc}$ represents the lifetime of ozone in the region where polar stratospheric clouds can potentially occur.

For the definition of $\tau_{psc}$, we are following Sinnhuber et al. (2003):

$$
\tau_{psc} =
\begin{cases}
10 \text{ days} & \text{for } \vartheta < 92.5° \text{ and } T < 195\text{K} \\
\infty \text{ days} & \text{else}
\end{cases}
\tag{2}
$$

with $\vartheta$ as the solar zenith angle.





## 4.2 Vortex tracer

In the region of the polar stratospheric vortex, temperatures below 195 K can be observed. This low temperature regime
determines the development of polar stratospheric clouds. Chlorine and Bromine activation takes place on the surfaces of polar
stratospheric cloud particles. This activation leads to ozone destruction. In the past, it has been observed that air parcels within
the polar vortex seem to be encapsulated (Loewenstein et al., 1989; Russell et al., 1993), because the polar vortex edge can act
as a transport barrier.

   To investigate processes which are relevant in the region of the stratospheric polar vortex, we introduce a polar vortex tracer
in ICON-ART. The processes of interest include transport processes of air parcels from within the vortex, exchange processes
at the vortex edge and mixing processes. The vortex tracer is a passive tracer with no destruction over time. This means that
the tracer is only affected by transport tendencies and has no interaction with other tracers or the dynamics. To define the edge
of the polar vortex, the first approach is to use Ertel's Potential Vorticity (PV) which is defined as followed:

$$PV = \frac{\zeta + f}{\rho} \frac{\partial \theta}{\partial z} \tag{3}$$

with $\zeta$ the relative vorticity, $f$ the Coriolis parameter, $\rho$ the air density, $\theta$ the potential temperature and $z$ the geometric height.
On a timescale of weeks and on an isentropic surface, the PV is a conserved quantity (Nash et al., 1996). The maximum in
the PV gradient defines the polar vortex edge. However, this metric shows high variations in small horizontal regions, distinct
maxima cannot always be defined. Therefore, we use a second metric following Nash et al. (1996). At the polar vortex edge,
the horizontal (zonal) wind component maximises and provides a surrogate for the PV gradient. In the model, the following
steps are performed separately for each hemisphere: first, the PV is calculated at every grid point. Between minima and maxima
of PV, a sufficient number of equidistant intervals is defined. Then, the respective geographical area enclosed by PV isolines
is calculated. By determination of maximum westerly wind relative to the area under PV isolines and multiplication of both
values, is given a reasonable constraint for the polar vortex edge boundary region. The described definition holds for the
location of the northern hemisphere polar vortex. For the southern hemisphere, the maximum of the zonal wind gives the
constraint for the location of the edge.

   At the initialisation time step, the passive tracer is initialised to one within the boundaries of the vortex. Afterwards only
transport processes are in action. This provides a useful metric to investigate transport processes in the region of the polar
vortex boundary.

## 4.3 Age of air

The characterisation of stratospheric transport from observations is difficult, because the velocity of the meridional overturning
(the so-called Brewer-Dobson circulation) cannot be accessed directly. However, the efficiency of the overturning can be
estimated by investigating the trends of volume mixing ratios of some long lived trace gases (e.g. Rosenlof, 1995; Bönisch
et al., 2011). One of these species is sulfur hexafluoride ($SF_6$), which is an anthropogenically emitted chemical compound
with a long life time of up to thousands of years (e.g. Ray et al., 2017). The increase of the middle atmosphere mixing ratio





of $SF_6$ can be assumed as quasi-linear (Maiss and Levin, 1994). In addition, $SF_6$ has no significant chemical sink within the stratosphere. In the mesosphere, photolytic dissociation at the Lyman-$\alpha$ band and dissociative electron detachment should be taken into account (Ravishankara et al., 1993). With some assumptions, it is possible to calculate the age of air by using the $SF_6$ concentration at a specific layer and horizontal location and comparing (matching) it to past tropical tropospheric values. Thus, the age of air represents the time that an air parcel needed to travel from the tropical tropopause to this particular location in the stratosphere. In the model a simplified procedure is followed: Lower boundary conditions are updated every integration time step to simulate a linear increase of $SF_6$ with time. The increase corresponds to an increase of the value of one year per year. Using the time lag technique (e.g. Schmidt and Khedim, 1991; Reddmann et al., 2001), one can calculate the age of air at this point by:

$$\Gamma_{\text{age}} = 7 \cdot 86400 \cdot 365.2425 + \Delta t_{\text{sim}} \tag{4}$$

with $\Delta t_{\text{sim}}$ is the integration time step of the model given in s. After initialisation, the tracer $\psi_{\text{age}}$ is transported. To neglect deviation from the initial assumption of a strictly linear growth of $SF_6$, the mean age of air is taken into account for further analysis. For the final calculation, the simulated values are merged with the simulated time to get the actual age of air in years ($\psi'_{\text{age}}$):

$$\psi'_{\text{age}} = \frac{t^\star_{\text{sim}} - (\frac{\psi_{\text{age}}}{86400} - 7 \cdot 365.2425) - t^\star_{\text{init}}}{365.2425} \tag{5}$$

where $t^\star_{\text{sim}}$ is the simulated time, $t^\star_{\text{init}}$ the time of initialisation, both given in the representation of the Julian Day.

### 4.4 Water vapour

Water vapour has a strong impact on the radiation budget of the atmosphere in the long-wave infrared spectrum. The infrared emission of water vapour leads to a cooling in the thermal budget of the atmosphere. Thus, the concentration of water vapour influences transport processes due to thermodynamic induced changes in the wind field. On the other hand, transport processes affect water vapour concentrations. The investigation of simulated water vapour distributions allows the study of the global circulation in models and observations (Kley et al., 2000). The amount of water vapour entering the stratosphere depends on the temperature within the tropical tropopause layer (TTL). Above, the Brewer-Dobson circulation drives the upward transport in the tropics. Below, convective fluxes and slow ascend in the TTL determine vertical transport. The process of "freeze-drying" leads to dry air parcels entering the stratosphere, because ice crystals sediment out (Brasseur and Solomon, 2006). This process is included in the microphysical schemes of the NWP and the climate physics. Methane oxidation is a chemical source for stratospheric water vapour. The photodissociation, mainly located in the mesosphere, is an important sink for water vapour in the atmosphere.

We are following Dethof (2003) by using the parameterisation:





$$
\tau_{\mathrm{CH4}} = \begin{cases} 100 & p \leq 50\,\mathrm{Pa} \\ 100 + \alpha\,\dfrac{\ln\left(\frac{p}{50}\right)^4}{\ln\left(\frac{10000}{p}\right)} & 50\,\mathrm{Pa} < p < 10000\,\mathrm{Pa} \\ \infty & p \geq 10000\,\mathrm{Pa} \end{cases}
\tag{6}
$$

with $\tau_{\mathrm{CH4}}$ the life time of methane in seconds, $p$ the pressure , and $\alpha = \frac{19\ln(10)}{\ln(20)^4}$. Based on Brasseur and Solomon (2006), the lifetime of water vapour $\tau_{\mathrm{H2O}}$ due to photodissociation can be calculated by:

$$
\tau_{\mathrm{H2O}} = \begin{cases} 3 & p \leq 0.1\,\mathrm{Pa} \\ 100 + 1 + \alpha\,\dfrac{\ln\left(\frac{p}{50}\right)^4}{\ln\left(\frac{10000}{p}\right)} & 0.1\,\mathrm{Pa} < p < 20\,\mathrm{Pa} \\ \infty & p \geq 20\,\mathrm{Pa} \end{cases}
\tag{7}
$$

Taking both terms of production and loss into account, the volume mixing ratio of water vapour at time step $t + \Delta t_{\mathrm{sim}}$ can be calculated by

$$
\psi_{\mathrm{H_2O}}(t + \Delta t) = \psi_{\mathrm{H_2O}}(t) + \Delta t \left( 2\frac{1}{\tau_{\mathrm{CH_4}}} \psi_{\mathrm{CH_4}}(t) - \frac{1}{\tau_{\mathrm{H_2O}}} \psi_{\mathrm{H_2O}}(t) \right)
\tag{8}
$$

with $\psi_{\mathrm{CH_4}}$ as the methane tracer of ICON-ART.

Studies have shown that observed mean-annual cooling trends in the tropical tropopause are larger than shown by model
simulations, e.g. Shine et al. (2003). It can be seen that the large scale dynamics in the Earth's atmosphere, tropical tropopause temperatures, and lower stratospheric water vapour are closely linked to each other by complex feedback processes. By developing an extension of ICON-ART, we want to account for these mechanisms in a simple but reasonable way. To investigate the feedback processes of the ART water vapour tracer on, e.g. radiation, only the tendency from the methane oxidation and photodissociation are taken into account. At every model time step, the water vapour mass mixing ratio is set to the value of
the $q_v$ tracer, which is affected by the micro physics routines of ICON. The $q_v$ tracer is the standard water vapour tracer of ICON. Within the ICON-ART routine, the tendency due to methane oxidation and photodissociation can be added. In the last step, $q_v$ is set to the value of $\psi_{\mathrm{H_2O}}$.

## 5   Applications

Here, we use ICON-ART and its new tracer framework with the NWP and the climate physics configuration in different
applications. First, we discuss a stratospheric hindcast experiment based on the ICON NWP configuration, focusing on different chemistries (Section 5.1). Second, we show some climatological applications based on the ICON climate physics configuration, illustrating the impact of composition feedbacks.



## 5.1 Ozone and Vortex tracer

In 2002, an unusual split of the ozone hole was observed on 22 September 2002 and described by Newman and Nash (2005). The initiation of the splitting process is not fully understood yet. The split of the ozone hole had no chemical reason; instead it is dynamical change controlling composition and in particular ozone distributions. Several studies, e.g. Matsuno (1971)

have shown that a vortex split event can be caused by atmospheric interactions with upward propagating planetary waves. Sinnhuber et al. (2003) pointed out, that the major stratospheric warming occurred far earlier than the normal final warming at the end of the ozone hole season. With ICON-ART we are able to study the vortex split event of 2002 in a hindcast. First, we want to discuss the total ozone column simulated with ICON-ART. For the setup of the experiment, we use reanalysis data from the European Centre for Medium-Range Weather Forecasts (ECMWF) - ERA-Interim (Dee et al., 2011) to initialise the

meteorological variables (e.g. pressure, temperature, water vapour and horizontal wind fields). The hindcast is initialised on 20 September 2002 00:00 UTC. The chosen grid is R2B6, corresponding to an approximate horizontal grid spacing of $40\,\mathrm{km}$. The model top is at $75\,\mathrm{km}$ with 90 vertical levels. The integration time step is $240\,\mathrm{s}$, and the output time step every hour. The simulated ozone column, interpolated on a regular latitude - longitude grid with a resolution of $0.5°$, is shown in Figure 3. The four columns show daily means of total ozone for the respective dates. The mean is taken over all 24 output steps starting

at 00:00 UTC. For the initial values of ozone, the ERA-Interim data is used analogous to the meteorological data. The ozone hindcasts are performed with two different schemes: The modified Linoz scheme as described in section 4.1 and a gasphase algorithm (the extended Chapman cycle) with reactions described in Table 2.4. For the comparison between simulations and measurements, we are using satellite observations from TOMS instrument (TOMS Science Team, 2016).

Five days after initialisation, the represented horizontal geometry of the total ozone column in both ICON-ART simulations

is slightly different in comparison to the satellite observations. After more than ten days of simulation, the shape of the vortex is in good agreement with satellite observations again. Within the polar vortex, the total ozone column reaches values of about 200 DU in both, ICON-ART simulations and the satellite observations. At the end of the simulation, on 1 October 2002, the largest difference between the extended Chapman cycle simulation, using the full gas phase algorithm, and the simplified Linoz simulation is found at $120°$W, $70°$S, outside of the polar vortex. The total ozone column reaches values of 450 DU and above

for the extended Chapman cycle simulation. For the Linoz simulation, ozone destruction is slightly higher and values up to 400 DU are reached.

In order to illustrate the differences between the two idealised simulations. One is parameterised, Linoz; the other considers gas phase chemistry only, extended Chapman cycle, the total contribution of chemical tendencies is depicted in Figure 4. We are using the differences between a passive (no chemical changes after initialisation) ozone tracer and the chemical ozone

tracers for Linoz and extended Chapman cycle chemistry. The difference represents the chemical ozone loss. Note that for visualisation purposes, values for the ozone losses in Figure 4 are scaled by a factor of 10.

The passive and the chemical ozone tracers are initialised identically. Technically this is ensured by declaring the ozone tracers to be of the same type as the chemical tracer, but without chemical interactions. The corresponding XML entry can be found in Appendix A1. In Figure 4 at $120°$W, $70°$S, on 1 October 2010, ozone loss is dominated by chemical loss. Ozone





losses are positive in that region: The passive tracer, which is only affected by dynamical tendencies, has higher values than the chemical tracers, thus ozone has been depleted by the chemistries. The maximum difference between passive and chemical tracers is up to ten times higher for the Linoz simulation than for the extended Chapman cycle. This is to be expected, because we do not consider additional heterogeneous processes in the extended Chapman cycle chemistry. However, we can focus on

the general spatial structures and how transport has modified ozone distributions. Inside the polar vortex, on 1 October 2002 , we model negative ozone loss for both simulations. The chemical tracer in both simulations is increased with respect to the passive one. The increase is higher in the Linoz simulation than in the extended Chapman cycle. This implies that temperatures in that region are not low enough to trigger the heterogeneous destruction of ozone in the Linoz scheme. Outside the polar vortex, mainly on 25 September, high values of ozone loss can be observed for the Linoz simulation but not for the extended

Chapman cycle. This is also caused by the difference in addressing heterogeneous destruction. Within the Linoz scheme, the loss term has been triggered and we can observe additional ozone loss. This feature is missing for the extended Chapman cycle chemistry.

In Figure 5, the temporal evolution of the passive vortex tracer is depicted. The colour coding gives the values of the vortex tracer at an interpolated pressure level of $30\,\mathrm{hPa}$ at midnight of the given date. Again, the date of initialisation is chosen

to be 20 September 2002 and within the boundaries of the vortex, the tracer is filled with values of one. In the hindcast, only transport takes place. The horizontal spreading of the vortex tracer depicts the dynamical evolution of the vortex in the southern hemisphere. The horizontal spreading and steep tracer gradients correspond to the horizontal distribution of the total ozone column in Figure 3. At the day of initialisation, the vortex is still intact, but in the following days the first observed major stratospheric warming in southern hemisphere e.g. (Newman and Nash, 2005) occurs. The massive outflow of vortex air

masses, beginning on 24 September 2002, can be visualised by the vortex tracer distributions. This outflow is correlated to the increased dynamical impact on the vortex integrity. The vortex split occurred on 26 September 2002. The structure, represented by the spatial distribution of the vortex tracer, is nearly separated. On this day, the northern-most latitude of $30°$ S is reached by a vortex filament. The vortex tracer allows to define regions of isolated air masses within the vortex, thus providing an insight into the chemical composition changes that are least affected by diffusion and mixing, e.g. (McKenna et al., 2002) or (Konopka

et al., 2005).

## 5.2    Feedback of ozone on radiation

In the previous section we have shown that the Linoz configuration of ICON-ART provides good hindcasts on days to weeks. Here, we will extend the time horizon considered to decadal integrations. In addition, we will illustrate how the optional composition of tracers with radiation feedback affects the system. With the flexible tracer structure the user gets the ability to

switch on the radiative feedback by the tag

```
<feedback> 1 </feedback>.
```

Thus, no changes in the code have to be made by the user. Only the XML file has to be changed.

The ICON-ART simulation is configured as an AMIP (Atmospheric Model Intercomparison Project) like experiment (Gates et al., 1999). The boundary conditions used are summarised in Table 2. We set up two experiments on the R2B4 grid which





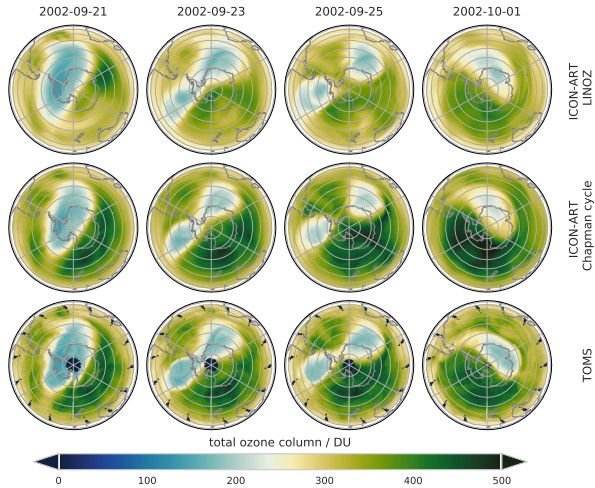

**Figure 3.** Antarctic total ozone column [DU] for a time sequence starting on 21 September 2002 finishing 01 October 2002 (left to right). Daily averages for two ICON-ART simulations for the Linoz chemistry scheme (top) and for the extended Chapman cycle chemistry (middle). TOMS observations are shown in the bottom. The model was initialised on 20 September 2002.

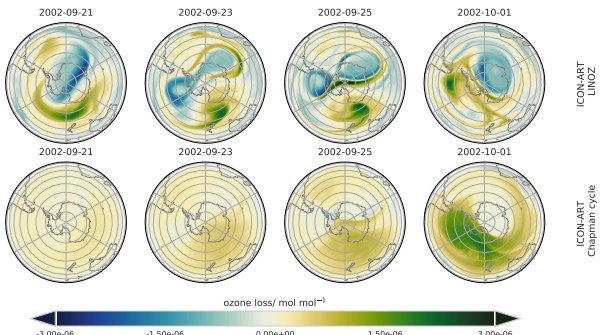

**Figure 4.** Difference of the passive and active ozone tracers [mol mol$^{-1}$] at 50 hPa over Antarctica. Daily averages for two ICON-ART simulations for the Linoz chemistry scheme (top) and for the extended Chapman cycle chemistry (bottom). The loss rate for the extended Chapman cycle is multiplied by a factor of ten for the visualisation.

corresponds to an effective horizontal grid spacing of 160 km. The integration time step is chosen to be 600 s. Output is written every three days and interpolated onto pre-defined pressure levels, corresponding to the standard ERA-Interim pressure levels. The first experiment uses an ozone climatology with monthly mean values based on Cionni et al. (2011). This simulation is called control. The feedback simulation uses the interactive ozone as well as the additional tendencies on water vapour, shown

5 in Section 4.4. For all following diagnostics and discussions, the same two simulations are used.

Figure 6 shows the climatological ozone distribution at a pressure level of 50 hPa. Monthly averaged zonal means of ozone for the period of 1980 to 2009 are plotted twice. The left panel shows the result of the control run (ctrl) without any feedback





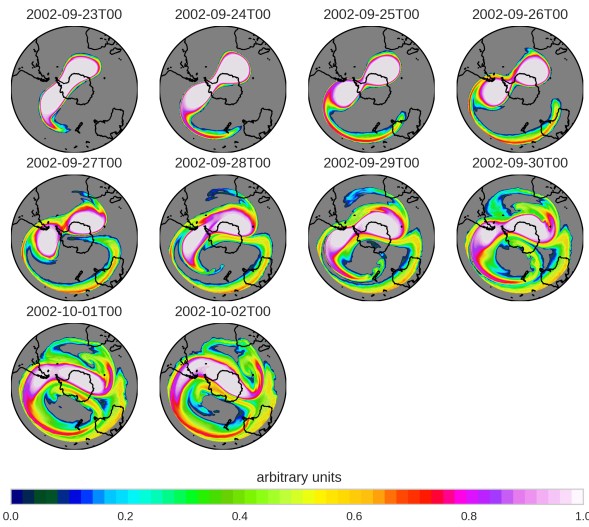

**Figure 5.** Daily snapshots at midnight UTC at 30hPa of the passive vortex tracer in arbitrary units (upper left to bottom right). The passive tracer was set to one within the vortex boundary at the beginning of the integration.

**Table 2.** Overview of the boundary conditions used in the AMIP like experiments.

| Variable | Reference |
|---|---|
| SST/SIC | Taylor et al. (2000) |
| Spectral solar irradiation | Lean et al. (2005) |
| greenhouse gases RCP 4.5 | Riahi et al. (2007) |
| O3 concentration | Cionni et al. (2011) |
| tropospheric aerosol | Stenchikov et al. (1998, 2004, 2009) |
| stratospheric aerosol | Stenchikov et al. (1998, 2004, 2009) |

from the modelled ozone distribution. Instead, the ozone climatology as described in Table 2 is represented. The middle panel shows the result of the non-interactive Linoz simulation. Thus, modelled temperatures are the same as in the control run. The right panel shows the temporal evolution of the zonal mean ozone in the feedback simulation. Additionally, contour lines are representing the inter annual standard deviations of ozone in all panels. The black contour line represents a standard deviation

5   of $1 \times 10^{-7} \text{kg kg}^{-1}$. Brighter colours present higher values of standard deviation with a spacing of $2 \times 10^{-7} \text{kg kg}^{-1}$. The mean year from January to December is plotted twice, for a better representation of the temporal evolution.

A striking difference in the three ozone distributions shown in Figure 6 is the duration of the ozone hole period in dark blue shadings. Modelled ozone (middle and right panel) shows a much longer duration than the assumed background climatology. In addition, the duration of the ozone hole period increases slightly for the interactive integration. This is caused by the feedback





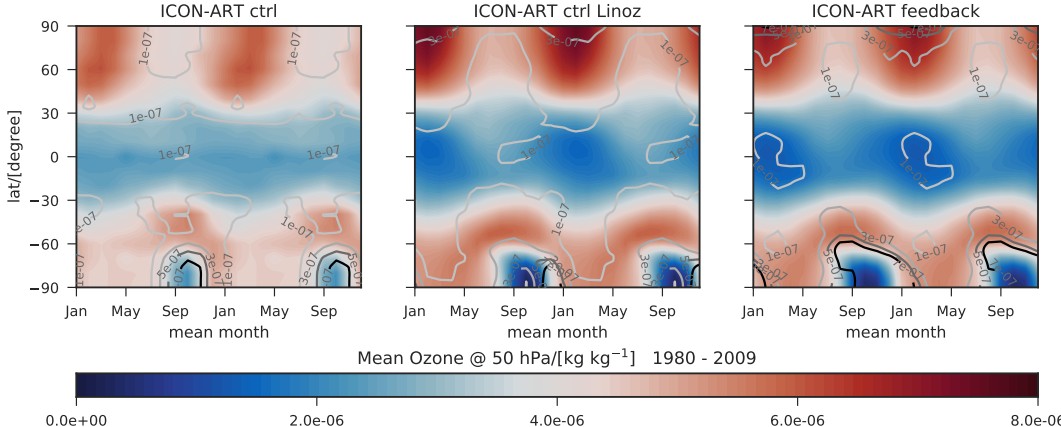

**Figure 6.** Monthly averaged zonal means of ozone $[\mathrm{kg\,kg^{-1}}]$ at 50 hPa (shown twice) for the period from 1980 to 2009 (shaded). Contour lines represent the standard deviation of the monthly means. Left panel: ozone climatology as used in the control simulation; middle panel: non-interactive Linoz ozone; right panel: interactive Linoz ozone.

of the modelled ozone, calculated with the Linoz parametrisation under consideration of the correction term for a shorter ozone life time due to the presence of polar stratospheric clouds. From October to December, very low ozone concentrations of about $1 \times 10^{-6} \mathrm{kg\,kg^{-1}}$ can be seen in the panel for the non-interactive Linoz ozone (middle panel). For the feedback simulation, low ozone concentrations in the southern hemisphere, in conjunction with low temperatures, occur until the end of January.

The feedback process stabilises the southern hemisphere polar vortex, prolonging its lifetime by delaying the final warming. The default climatology of the ICON-ART control simulation does not represent very low values of ozone concentrations. This misrepresentation has also been discussed by e.g. Arblaster et al. (2014). Here, the authors point out that most models that are using a prescribed ozone climatology tend to under estimate the Antarctic ozone depletion. Thus modelled ozone values are higher than indicated by observations between 1979 and 2007 (e.g. Hassler et al., 2013). Taking the standard deviation into
account, the characteristics of the AMIP ozone climatology (Cionni et al., 2011) becomes clearer. The contour lines represent a semicircle each winter in the southern hemisphere that is aligned with the ozone concentration gradients. This symmetric and coherent pattern of the ozone minimum is most likely not a very realistic representation of variability on top of an ozone hole that is not deep enough. The standard deviation isolines for the modelled ozone are different and intersect the isolines of ozone concentrations, with higher variabilities at later times in the ozone hole period. This seems a more realistic behaviour, however
we do not claim that this is already a perfect representation in comparison to the reanalysis data.

## 5.3 Temperature changes due to ozone feedback

In the previous section we have shown that with the change to an interactive representation of ozone the duration of the southern polar vortex is increased. Here, we provide more details of the zonal temperature distributions in the control and feedback simulations.





The direct impact of ozone, already mentioned in Section 5.2, is also displayed in the seasonal variation of the zonal mean temperatures.

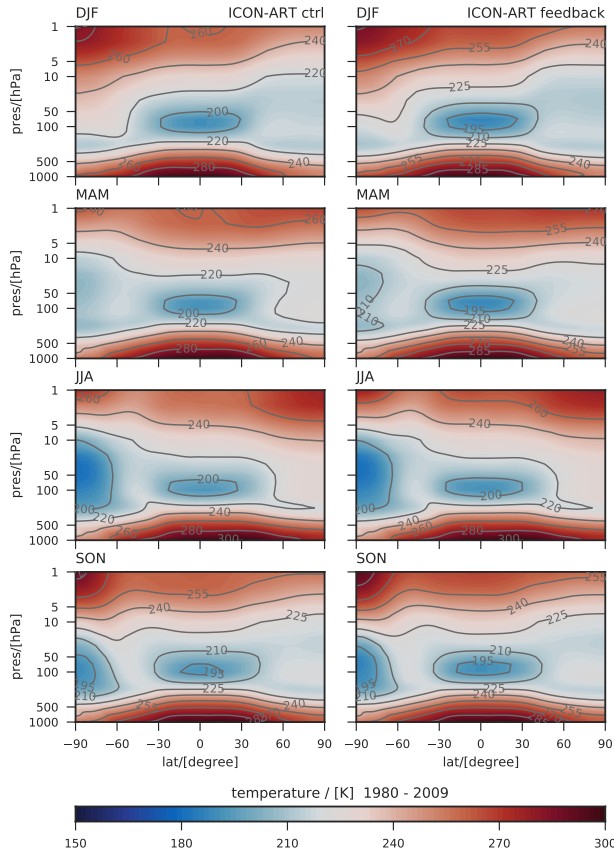

**Figure 7.** Latitude-height cross sections of seasonal and zonal mean temperature [K] for ICON-ART simulations from 1980 - 2009. Left panel: control run; right panel: feedback simulation.

In the northern hemisphere winter (DJF), temperatures of 210 K are reached between 100 hPa and 10 hPa, in the tropics. For the season of June-July-August (JJA), the temperature minimum in the tropics is at 100 hPa with temperatures as low as 200 K in both ICON-ART simulations. In the southern hemisphere winter, ICON-ART reaches temperatures of about 200 K. The northern hemisphere summer is represented by temperatures higher than 260 K above 5 hPa.

Figure 8 shows the difference between control and feedback run. In the southern hemisphere winter, the effect described in Section 5.2 can be observed. Due to the lower polar vortex temperature, differences up to 5 K occur. The control run shows warmer temperatures in the southern polar region (below 20 hPa). Above this altitude the feedback run is warmer than the control run. This is due to the different ozone distributions in the winter southern hemisphere. Within the tropical stratosphere, temperature differences of about 2.8 K can be seen. The differences of zonal and seasonal mean temperatures between ERA-Interim and the ICON-ART feedback simulation is shown in Figure 9. The feature of a long lasting southern hemisphere





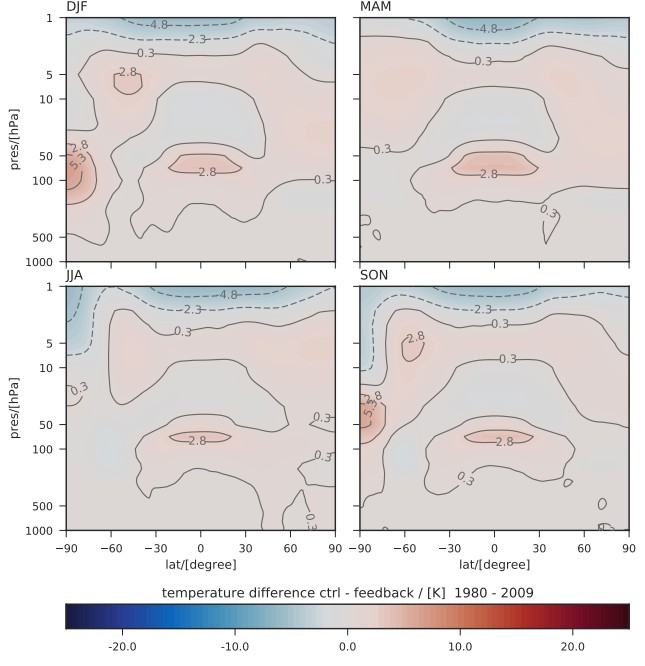

**Figure 8.** Latitude-height cross sections of seasonal and zonal mean temperature differences [K] for control minus feedback simulation, as shown in Figure 7.

polar vortex is present as well, seen in high temperature differences of up to 20 K from September to November (SON). In general, the ICON-ART control simulation shows warmer temperatures than the ICON-ART feedback simulation, except for high altitude ranges above 5hPa.

The general structure is comparable to the results shown in the comparison studies of ECHAM5 (Roeckner et al., 2006). The difference between ERA-Interim and ICON-ART is increased in the southern hemisphere stratosphere. Nevertheless, the representation of the polar vortex seems to be more realistic in the ICON-ART feedback simulation than in the control simulation. In the vertical region around $50\,\text{hPa}$, the difference between ERA-Interim and the feedback simulation is about $15\,\text{K}\,to\,20\,\text{K}$ in the tropics and below 8 K in the southern hemisphere.

## 5.4 Zonal wind fields

Changes in temperature due to radiative feedback effects of ozone are also affecting the zonal wind structure. The zonal mean zonal wind is shown in Figure 10. The left column shows the seasonal mean of the control ICON-ART simulation and the right column the feedback simulation. In both simulations, a strong eastward zonal wind with wind speeds up to $60\,\text{m}\,\text{s}^{-1}$ is reached in the southern hemisphere winter (JJA). The wind speed patterns in the tropical stratosphere also matches the seasonal mean analysis.





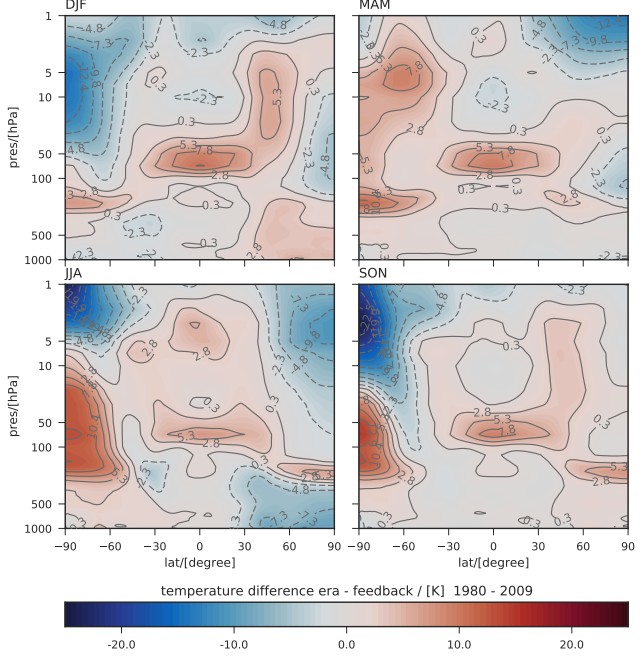

**Figure 9.** Latitude-height cross sections of seasonal and zonal mean temperature differences [K] for ERA-Interim minus feedback simulation.

Figure 11 shows the differences between the ERA-Interim and the results of the ICON-ART feedback simulation. Strong differences in the stratospheric zonal wind can be seen in the northern hemisphere winter. Here, the ERA-Interim shows values up to $25 \text{ m s}^{-1}$ higher than in the ICON-ART simulation. Between $30°$N and $60°$N latitude above $20 \text{ hPa}$ the sign of the differences changes. Here, we observe stronger zonal wind speeds than in ERA-Interim. The overall patterns are similar to the differences shown in Roeckner et al. (2006).

## 5.5 Water vapour

Here, we focus on the atmospheric water vapour tape recorder in ICON-ART. An atmospheric tape recorder can be defined as the vertical propagation of an anomaly that varies periodically in time with a tropospheric source (Gregory and West, 2002). The temporal and vertical distribution of the tropical stratospheric water vapour is a prominent example for an atmospheric tape recorder signal. The simulated stratospheric water vapour depends strongly on the temperatures that are encountered by an air parcel containing water vapour that is transported vertically from the troposphere upwards towards and through the tropopause (Schoeberl et al., 2012). The quantitative link between variations of tropical tropopause temperatures over decades and their influence on water vapour transfer into the stratosphere is still not fully understood (e.g. Rosenlof and Reid, 2008). It has been shown that stratospheric water vapour can have a strong impact on stratospheric climate (e.g. de F. Forster and Shine, 1999). Thus, the study of the water vapour tape recorder is an important tool for the further understanding of large scale transport processes and climate change linkages.





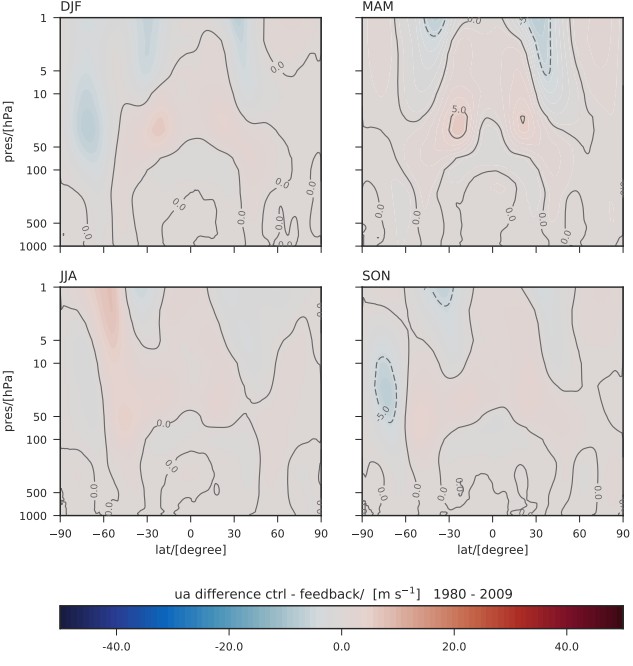

**Figure 10.** Latitude-height cross sections of seasonal and zonal mean zonal wind $[\mathrm{m\,s^{-1}}]$ for ICON-ART simulations from 1980 - 2009. Left panel: control run; right panel: feedback simulation.

The tape recorder is calculated from the annual mean anomalies of the water vapour tracer in the tropics ($5°N - 5°S$). For this study we use the water vapour tracer, $q_v$. This tracer is the standard tracer of ICON itself. It is used in the radiation scheme and is not only transported but is also affected by the microphysical schemes. As in the previous section, ozone calculated with the Linoz scheme was used interactively. The additional tendency by methane oxidation and photolysis is also included in the stratospheric water budget.

Figure 12 shows the mean stratospheric tape recorder for thirty years of simulation from 1980 to 2009. For analysis, the year of initialisation (1979) is excluded. This reduces any possible spin up effects. We compare the ICON-ART results with the water vapour product from ERA-Interim. The calculated mean ERA-Interim tape recorder is shown in the bottom panel of Figure 12.

The tape recorder signal for the ICON-ART control simulation shows lower values of anomalies down to $-7\times10^{-7}\,\mathrm{kg\,kg^{-1}}$ in comparison to the feedback simulation (Figure 12). For ERA-Interim, absolute values up to $-2\times10^{-7}\,\mathrm{kg\,kg^{-1}}$ are diagnosed from February to June in the pressure range of $100\,\mathrm{hPa}$ to $50\,\mathrm{hPa}$.

The anomalies in the feedback simulation show higher absolute values. The dry anomalies of the control simulation, between April and June, are decreased in the feedback simulation. In addition, the wet anomalies from June to December, between $60\,\mathrm{hPa}$ to $20\,\mathrm{hPa}$, are decreased in the feedback simulation. The water vapor tape recorder shows less pronounced dry anomalies in the tropical stratosphere in the feedback simulation due to the additional source of water vapour by methane



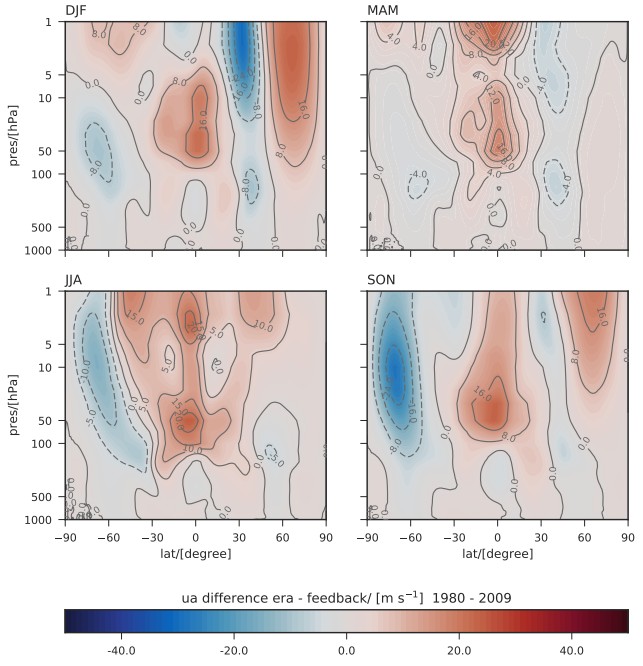

**Figure 11.** Latitude-height cross sections of seasonal and zonal mean zonal wind differences $[\mathrm{m\,s^{-1}}]$ for ERA-Interim minus feedback simulation.

oxidation. Additionally, the analysis of the water vapour tape recorder is consistent with the results of the temperature differences, as seen in Figure 8. Due to lower tropical tropopause temperatures in the feedback simulation, less water can enter the lower stratosphere. The results correspond with the freeze-drying hypothesis explained above. In general, monthly mean anomalies are attenuated in the feedback simulation compared to the simulation using the standard ozone climatology, with in

particular the winter months being more similar to ERA-Interim.

The slope of the height-altitude water vapour anomalies is nearly unchanged between non-interactive and interactive integrations. Thus, the velocity of the upward transport is largely unaffected by the inclusion of the radiative feedback. The result of both ICON-ART simulations in comparison to ERA-Interim is similar to the results presented in Jiang et al. (2015). Here, the authors combined measurements and simulations of water vapour from the Microwave Limb Sounder (MLS), GMAO

Modern-Era Retrospective Analysis for research and Applications in it's newest version (MERRA-2) and ERA-Interim and used them for a comparison of the water vapour tape recorder behaviour. The upward transport above the tropical tropopause of ERA-Interim is found to be faster than the transport diagnosed from MLS measurements. Thus, in the current configuration ICON-ART produces similar ascend rates to ERA-Interim, which are likely too fast.

We have shown that the change between interactive and non-interactive integrations with respect to tropical ascend rates is

15 small. However, some changes are clearly detectable and the important relationship between the tropical tropopause minimum





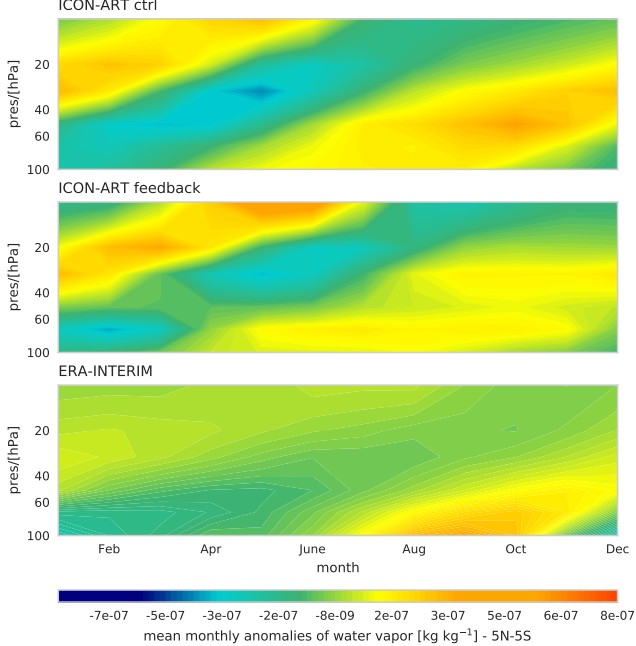

**Figure 12.** Tropical (5°N-5°S) water vapour anomalies as monthly mean deviations from the annual mean, averaged from 1980 to 2009, as a function of months and altitude.

temperatures and water vapour concentrations is qualitatively captured by the ICON-ART system. More comprehensive climate studies are in preparation.

## 5.6   Age of air

For the simulation of the age of air we use the same setup as described in section 5.3. As all other diagnostics, the age of air tracer is interpolated on a regular latitude-longitude grid with a horizontal resolution of 0.75 ° × 0.75 ° on predefined pressure levels.

   The tracer is initialised as described in section 4.3. The control (ctrl) experiment is a simulation in which water vapour and ozone calculated within ICON-ART have no impact on the calculation of radiation. In the second experiment (feedback), the altered ozone and water vapour distributions of ICON-ART are coupled to the radiation routine. The first eleven years of each simulation are excluded in the analysis to prevent spin-up effects contaminating the result.

   The ICON-ART modelled age of air is depicted in Figure 13. The diagnostic of age of air can be seen in this case study as an important tool to analyse the feedback processes of greenhouse gases on transport processes in the Earth's atmosphere. It can be seen that the mean age of air is younger, thus the upward transport has been faster in the control simulation. Due to upwelling transport processes in the tropics, the youngest air masses can be found there. The polar regions show the occurrence of older air masses up to an age of four years in both simulations. The asymmetry between southern and northern hemisphere,



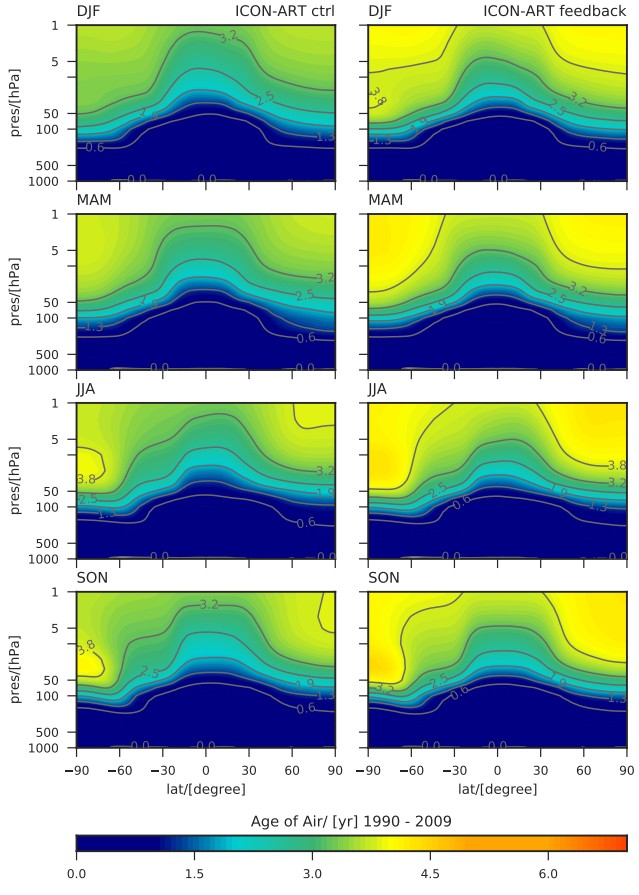

**Figure 13.** Latitude-height cross sections of seasonal and zonal mean age of air [year] for ICON-ART simulations from 1990 - 2009. Left panel: control run; right panel: feedback simulation.

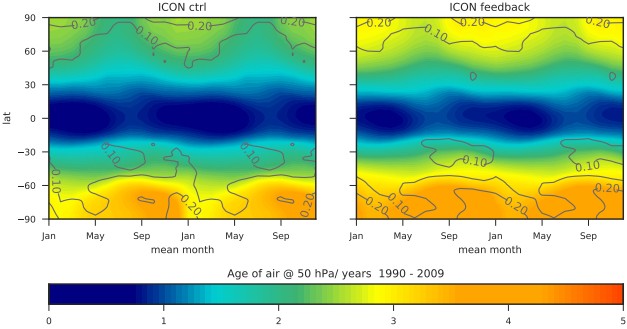

**Figure 14.** Monthly averaged zonal means of age of air [year] at 50 hPa (shown twice) for the period from 1990 to 2009 (shaded). Contour lines represent the standard deviation of the monthly means. Left panel: control run; right panel: feedback simulation.





induced by faster circulations in the southern hemisphere (e.g. Mahieu et al., 2014), is also well captured. One can clearly see the impact of ozone and water vapour on radiation and thus on transport processes. The age of air in the feedback simulation is up to six months older than of the control simulation. Figure 14 shows the climatological mean age of air, in the same representation as shown in Figure 6. Here, the temporal and zonal mean of the age of air from 1990 to 2009 is taken at an

altitude of $50\,\mathrm{hPa}$. The standard deviation from the mean is represented by the contour lines. These lines represent the inter-annual variability. The absolute mean age of air is higher for the feedback simulation on both hemispheres. The band of low values in the tropics is narrowed for the feedback simulation. The values of standard deviation are comparable. However, in the region of the southern hemisphere polar vortex, from October to January the standard deviation is higher for the feedback than for the control simulation. Since the polar vortex is stabilized by the ozone feedback, a different dynamical situation can

be observed, influencing the inter-annual variability of the age of air. In comparison to other studies, which focus on other time spans (e.g. Brasseur and Solomon, 2006; Engel et al., 2009; Stiller et al., 2012; Haenel et al., 2015), ICON-ART shows an age of air which is too young compared to observations. But this behaviour has also been observed in other studies with different models, as described in e.g. Monge-Sanz et al. (2007) or Hoppe et al. (2014). With this diagnostic the general representation of stratospheric transport processes can be further investigated.

## 6 Conclusions

We present a new flexible tracer framework developed for ICON-ART. The next generation model ICON-ART can be used for many different applications, currently ranging from forecasting to climate simulations. ICON is used for LES simulations, operationally for numerical weather forecasting, and for climate simulations. All three application areas have very different demands in terms of model configurations, including the set of tracers and tracer interactions to be simulated. For future

studies using ICON-ART, a fast adaption of the selected tracers and interactions to the experimental requirements is of high importance. With our new flexible tracer framework, tracers can be added and configured without any changes to the model source code. This allows users to easily perform complex model experiments. In a forward thinking manner, we also provide the option to extend the existing tracer structure and submodule awareness of tracer subsets. Within the scope of the paper, we demonstrate the tracer framework and its applicability for a range of simulations. We present one hindcast case study and

AMIP type climate integrations.

In the first instance we included a parameterised chemistry (Linoz) and a gas phase chemistry (extended Chapman cycle) into the NWP configuration of ICON-ART. With this setup we perform a successful hindcast experiment of the ozone hole split in the year 2002 and characterise the (chemical) ozone changes during the hindcats period. Using a diagnostic vortex tracer, we can identify the vortex remnants that are least influenced by chemistry. Results are consistent with expectations and previous

work and illustrate the ability of the ICON model to produce a good stratosphere forecast on timescales of days to weeks. In this situation dynamics is driving the ozone distribution and the spatial patterns correspond well to observations. However, ozone amounts differ between the two different chemistries. The setup can be easily extended by more chemical reactions and diagnostic tracers.

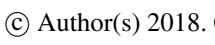



In the second instance we include a parameterised chemistry (Linoz), a methane oxidation scheme and a diagnostic tracer (age of air) into a climate configuration of ICON-ART and perform AMIP type integrations. We perform decadal non-interactive and interactive integrations and compare the performance of both simulations with each other and with ERA-Interim. For the interactive integration we couple ozone and water vapour to the radiation. In the interactive simulation the

ozone hole season is extended, the tropical upwelling is only weakly affected and the overturning circulation as measured by the Brewer-Dobson Circulation shows a northern hemisphere age increase. The base climatologies of ICON is not affected by ART in the non-interactive simulation. In the interactive simulation some changes to the climatologies of temperatures and winds occur, none of the changes are detrimental to the model, some are even beneficial.

For all experiments, no changes in the ART source code were necessary to change from NWP simulations to climate integra-

tion. Only the XML file differs between the full gasphase (extended Chapman cycle) simulation and the parameterised (Linoz) one. This paper demonstrates the flexibility of the new tracer framework for ICON-ART, which suites the demands of a large variety of different applications ranging from NWP to climate integrations.

*Code availability.* Currently the legal departments of the Max Planck Institute for Meteorology (MPI-M) and the DWD are finalising the ICON licence. If you want to obtain ICON-ART you will first need to sign an institutional ICON licence, which you will get by sending a

request to icon@dwd.de. In a second step you will get the ART licence by contacting Bernhard Vogel (bernhard.vogel@kit.edu). Versions are controlled by GIT repositories and a tar-ball of the latest official release is provided to the licensee.

*Competing interests.* There are no competing interests to declare.

*Acknowledgements.* This work was performed on the computational resource ForHLR II funded by the Ministry of Science, Research and the Arts Baden-Württemberg and DFG ("Deutsche Forschungsgemeinschaft").



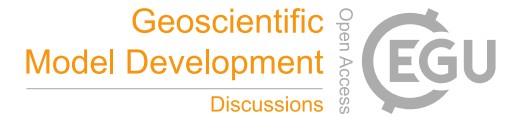

## Appendix A: XML files used in shown experiments

### A1  Ozone and Vortex tracer - NWP experiment

Passive tracers

```
<?xml version="1.0" encoding="UTF-8"?> <!DOCTYPE tracers SYSTEM "tracers.dtd">

       <tracers>

        <passive id="TR_vortex">

         <mol_weight type="real">1.000E+0</mol_weight>

          <lifetime type="real">25920000</lifetime>

<transport type="char"> ON </transport>

          <unit type="char">none</unit>

         </passive>

      </tracers>
```

Chemical tracers - Linoz

```
<?xml version="1.0" encoding="UTF-8"?>

     <!DOCTYPE tracers SYSTEM "tracers.dtd">

       <tracers>

         <chemical id="TRO3">

          <tag001 type="char">chemtr</tag001>

<mol_weight type="real">4.800E-2</mol_weight>

          <lifetime type="real">259200</lifetime>

          <transport type="char"> stdchem </transport>

          <init_mode type="int"> 1 </init _mode>

           <init_name type="char">O3</init_name>

<unit type="char">mol mol-1</unit>

          <lfeedback type="int"> 0 </lfeedback>

         </chemical>

         <chemical id="TRO3_pas">

          <tag001 type="char">chemtr</tag001>

<mol_weight type="real">4.800E-2</mol_weight>

          <lifetime type="real">2592000000000</lifetime>

          <transport type="char"> stdchem </transport>

          <init_mode type="int"> 1 </init _mode>

         <init_name type="char">O3</init_name>
```



```
        <unit type="char">mol mol-1</unit>

        <lfeedback type="int"> 0 </lfeedback>

      </chemical>

    </tracers>
```

Selection of chemical tracers for the extended Chapman Mechanism. Not all tracers are shown.

```
     <?xml version="1.0" encoding="UTF-8"?>
     <!DOCTYPE tracers SYSTEM "tracers_gp.dtd">
<tracers>
       <chemical id="N2O">
         <tag001 type="char">full</tag001>
         <mol_weight type="real">44.02E-3</mol_weight>
         <transport type="char">stdchem</transport>
<number type="int"> 1 </number>
         <init_mode type="int">0</init_mode>
         <unit type="char">mol mol-1</unit>
         <init_type type="char">N2O </init_name>
       </chemical>
<chemical id="N2O5" >
         <tag001 type="char">full</tag001>
         <mol_weight type="real">108.02E-3</mol_weight>
         <transport type="char">stdchem</transport>
         <number type="int"> 2 </number>
<init_mode type="int">0</init_mode>
         <unit type="char">mol mol-1</unit>
         <init _ype="char">N2O5 </init_name>
       </chemical>
       <chemical id="HO2">
<tag001 type="char">full</tag001>
         <mol_weight type="real">33E-3</mol_weight>
         <transport type="char">stdchem</transport>
         <number type="int"> 3 </number>
         <init_mode type="int">0</init_mode>
<unit type="char">mol mol-1</unit>
         <init_name type="char">HO2 </init_name>
       </chemical>
       <chemical id="H2O" >
         <tag001 type="char">full</tag001>
<mol_weight type="real">18E-3</mol_weight>
         <transport type="char">stdchem</transport>
         <number type="int"> 4 </number>
         <init_mode type="int">0</init_mode>
         <unit type="char">mol mol-1</unit>
<init_name type="char">H2O </init_name>
       </chemical>
       <chemical id="NO">
         <tag001 type="char">full</tag001>
         <mol_weight type="real">30.01E-3</mol_weight>
```



```
        <transport type="char">stdchem</transport>
        <number type="int"> 5 </number>
        <init_mode type="int">0</init_mode>
        <unit type="char">mol mol-1</unit>
<init_name type="char">NO </init_name>
      </chemical>
      <chemical id="NO3">
        <tag001 type="char">full</tag001>
        <mol_weight type="real">62.01E-3</mol_weight>
<transport type="char">stdchem</transport>
        <number type="int"> 6 </number>
        <init_mode type="int">0</init_mode>
        <unit type="char">mol mol-1</unit>
        <init_name type="char">NO3 </init_name>
</chemical>

    </tracers>
```





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
