# Peer review of "ICON-ART 2.1 - A flexible tracer framework and its application for composition studies in numerical weather forecasting and climate simulations"

_Geoscientific Model Development, 2017_

## Referee Comment (RC1) · Anonymous Referee #1 · 25 Mar 2018

The manuscript presents a number of unique features of the implementation of tracers in the ICON-ART model, including a flexible method of specifying tracers and their properties in a separate XML file, that is made possible by the use of object oriented programming in Fortran. In addition to the description of the implementation, results from a short simulation of the 2002 Antarctic vortex split using a simple stratospheric chemistry scheme and the Linoz parameterized ozone is presented. Results from a 30-year simulation are also discussed, comparing a simulation with interactive ozone using the Linoz parameterization with a focus on the impacts on dynamical variables

such as winds, temperature and age of air.

While I am sympathetic to the goals of model documentation supported by GMD, I find Sections 2.1 through 2.3, and particularly section 2.2.1, to be much too technically oriented – to the point of seemingly like a users guide rather than a scientific article. While I believe the discussion of particular aspects of object oriented programming in Fortran 2003 and the ways this can be linked with XML to control model configuration would find an audience of interested readers, the detailed discussion of the XML input files seems much too specific. I would strongly urge the authors to find a way to more generally discuss the way the model has been structured and the advantages you find using XML with Fortran 2003 and remove some of the specific examples of code that are included. As discussed in the particular comments below, there are also a number of areas where the discussion of the model results does not seem clear and this should be improved. With a few improvements the manuscript could provide a nice example of a modern approach to numerical model design and illustrate the flexibility of the modelling infrastructure with a couple of examples.

Page 2, Lines 2 – 7: Nothing scientific in this comment at all: the first paragraph has two completely separate ideas stuck together (non-hydrostatic and seamless prediction) and is a very difficult start for the reader.

Page 2, Lines 31-32: I might suggest changing the text from 'field, like large eddy simulations, numerical weather predictions and climate simulations ICON-ART can run with the different existing physics configurations.' to 'field, like large eddy simulations, numerical weather predictions or climate simulations, ICON-ART can run with different existing physics configurations.'

Page 4 Lines 6-7: Should the phrase '...not have any impact on other tracers of the (thermo-)dynamical of the simulated system.' be '... not have any impact on other tracers or the (thermo-)dynamics of the simulated system.'

Page 13, Lines 5-16: Age of air derived from a linearly increasing tracer is fairly widely

used and I do not think all the details are required of the implementation. But I would also note that equation (4) is a bit difficult to understand. Is the first term on the right-hand side (7 x 86400.0 x 365.2425) the initial value assigned to the age tracer everywhere in the model?

Page 15, Lines 5 – 12: Looking at Figure 4, there is considerable ozone production (negative loss) occurring in the Linoz simulation and the authors note this here: 'Inside the polar vortex, on 1 October 2002, we model negative ozone loss for both simulations. The chemical tracer in both simulations is increased with respect to the passive one. The increase is higher in the Linoz simulation than in the extended Chapman cycle. This implies that temperatures in that region are not low enough to trigger the heterogeneous destruction of ozone in the Linoz scheme. Outside the polar vortex, mainly on 25 September, high values of ozone loss can be observed for the Linoz simulation but not for the extended Chapman cycle. This is also caused by the difference in addressing heterogeneous destruction. Within the Linoz scheme, the loss term has been triggered and we can observe additional ozone loss. This feature is missing for the extended Chapman cycle chemistry.' From Figure 4, the production in the Chapman chemistry seems to be quite less, if not close to zero, but Figure 3 shows that total column ozone within the vortex in the Linoz simulation is actually considerably lower than in the Chapman simulation. I admit one is total column and the other is ozone at 50 hPa, but the differences in the chemical change in ozone at 50 hPa do not seem to agree with the differences in total column ozone in the two simulations. Can the authors provide some reasons why total column ozone inside the vortex is lower while chemical change at 50 hPa suggests it should be higher? I am also curious about the ozone loss that occurs outside the vortex in the Linoz simulation and that is mentioned in this section as well. Is it possible to add the 195K isotherm on to Figure 4 to help the reader unambiguously see where the PSC-chemistry parameterization is active? While I can imagine some filaments of the original vortex having temperatures below 195K, I am curious why the two lobes of the original vortex are not cold enough to activate the ozone loss parameterization.

Page 17, Lines 3-4: For the control simulation that uses specified ozone from Cionni et al. (2011) in the radiation, what water vapour is used for radiation? Is it the water vapour that crosses the tropopause, but with no addition from methane oxidation?

Page 19, Lines 9 – 14: The standard deviation of ozone is discussed here and the point that the variability in the Cionni et al. dataset shows a pattern very similar to the ozone contours themselves is made. The Cionni et al. ozone dataset was made using a standard climatological ozone with a temporal trend given by regression to an idealized reactive chlorine (Cly). I think what you are showing in the standard deviation is just the long-term trend in ozone itself – the evolution of the trend around the long-term mean for the 1980 – 2009 period. While the Linoz ozone actually has year-to-year variability but as it is constructed I don't think it has any long-term trend. This comparison does not seem very valid. And that brings up another point that since Linoz is designed for a particular time period, particularly the PSC parameterization, is it correct to use it over the full period from 1980 when chlorine loading will have evolved considerably? I might suggest restricting the comparison to 1990-2009 when the effects of ODSs are more fully realized.

Page 20, Line 5: 'In the southern hemisphere winter, ICON-ART reaches temperatures of about 200 K.' This seems to be like a significant warm bias that would have serious impacts on PSC processes. How does this statement agree with the findings shown in Figure 9 that the feedback simulation seems to be biased cold compared to ERA-Interim?

Page 20, Lines 7-8: Is the comparison reversed in the statement 'In the vertical region around 50 hPa, the difference between ERA-Interim and the feedback simulation is about 15 K to 20 K in the tropics and below 8 K in the southern hemisphere.'

Page 22, Lines 2-3: On the comparison of zonal winds to ERA-Interim the authors state 'Here, the ERA-Interim shows values up to 25 m s−1 higher than in the ICON-ART simulation. Between 30âŮę N and 60âŮę N latitude above 20 hPa the sign of
the differences changes. Here, we observe stronger zonal wind speeds than in ERA-Interim.' The finding of stronger zonal winds, particularly around the southern hemisphere stratospheric jet, does not seem to fit with the cold temperatures found the ICON-ART feedback simulation that are shown in Figure 9. I think the negative difference means the winds in ICON-ART are stronger.

Page 23 – the caption on Figure 10 does not seem to match with what is shown in Figure 10.

Page 24, Lines 2-3: It is difficult to see the difference in water vapour entering the stratosphere that is mentioned by 'Due to lower tropical tropopause temperatures in the feedback simulation, less water can enter the lower stratosphere.' because Figure 12 is the monthly anomaly to the annual mean. A quick mention of the difference in the annual mean in the lower stratosphere would help.

Page 25, Figure 12 – Do you have any idea why the maximum in the annual cycle in the feedback simulation around 80 hPa is so broad, stretching from May to October. The broad annual cycle in the bottom right corner of the feedback plot is also difficult to reconcile with the very narrow maximum in the top left corner.

---

## Editor Comment (EC1) · G. A. Folberth (Editor) · 6 Apr 2018

The ICON-ART code is hosted and version-controlled via a GIT repository. Access is available at any time but a license agreement needs to be signed before access can be granted (IP protection requirement). It has been agreed between authors and the editor that the editor obtains a license and will redistribute to the reviewers upon request.

---

## Referee Comment (RC2) · J. Bieser (Referee) · 2 May 2018

The comment was uploaded in the form of a supplement:
https://www.geosci-model-dev-discuss.net/gmd-2017-286/gmd-2017-286-RC2-supplement.pdf

This manuscript describes a novel framework for the implementation of reactive tracers into the ICON model. The framework takes advantage of the commonly used KPP software and implements it into the ICON model in a way that allows for the run-time implementation of complex chemical mechanisms. The presented work significantly enhances the current ICON-ART system. The implementation of additional state variables and associated chemical reactions into the ICON model requires a high level of programming expertise and poses an obstacle to its usage that should not be underestimated. The presented model development is an elegant solution that will allow a wide user range to implement different chemical reactions into the model.

Besides the description of the technical enhancements, the authors present a wide range of sample applications ranging from short term NWP calculations with a simple ozone chemistry to long term climate runs with life time bases chemical reactions. I have to say that although the manuscript is quite long I enjoyed reading it and can support publication in GMD.

However, there are several, mostly minor issues that need to be addressed:

1) My main complaint is that most of the evaluation is based on qualitative comparison. I am missing quantitative measures (e.g. bias, error). Especially in section 5.2 it would make sense to give the model bias for alternative model runs.

2) Make sure to explain all abbreviations, even those that might seem trivial.

**P1 L10:** AMPI

**P2 L13:** Here you need to introduce the abbreviation NWP. And it would also make sense to give the ECHAM abbreviation here.

**P10 Figure 2:** SSO

**P18 Table 2:** SST/SIC

3) Thoroughly check that all values are given with a unit

4) I suggest to combine Figures 8 & 9 as well as Figures 10 & 11.

5) Minor issues:

**P2 L5:** … the same dynamical core ….

As you know (and state later in the text) this is not the case for ICON (and I am not sure which other model has actually reached that ideal).

**P2 L22:** Here you should mention that the development is based on COSMO-ART. Maybe I am wrong but people do know COSMO-ART. In this case add a few sentences to clarify any differences between the ART in COSMO-ART and ICON-ART

**Fig. 1.**

**Supplement:**

This manuscript describes a novel framework for the implementation of reactive tracers into the ICON model. The framework takes advantage of the commonly used KPP software and implements it into the ICON model in a way that allows for the run-time implementation of complex chemical mechanisms. The presented work significantly enhances the current ICON-ART system. The implementation of additional state variables and associated chemical reactions into the ICON model requires a high level of programming expertise and poses an obstacle to its usage that should not be underestimated. The presented model development is an elegant solution that will allow a wide user range to implement different chemical reactions into the model.

Besides the description of the technical enhancements, the authors present a wide range of sample applications ranging from short term NWP calculations with a simple ozone chemistry to long term climate runs with life time bases chemical reactions. I have to say that although the manuscript is quite long I enjoyed reading it and can support publication in GMD.

However, there are several, mostly minor issues that need to be addressed:

1) My main complaint is that most of the evaluation is based on qualitative comparison. I am missing quantitative measures (e.g. bias, error). Especially in section 5.2 it would make sense to give the model bias for alternative model runs.

2) Make sure to explain all abbreviations, even those that might seem trivial.

**P1 L10:** AMPI

**P2 L13:** Here you need to introduce the abbreviation NWP. And it would also make sense to give the ECHAM abbreviation here.

**P10 Figure 2:** SSO

**P18 Table 2:** SST/SIC

3) Thoroughly check that all values are given with a unit

4) I suggest to combine Figures 8 & 9 as well as Figures 10 & 11.

5) Minor issues:

**P2 L5:** … the same dynamical core ….

As you know (and state later in the text) this is not the case for ICON (and I am not sure which other model has actually reached that ideal).

**P2 L22:** Here you should mention that the development is based on COSMO-ART. Maybe I am wrong but people do know COSMO-ART. In this case add a few sentences to clarify any differences between the ART in COSMO-ART and ICON-ART

**P7 L6:** Here I got lost: prognostic and diagnostic state. Maybe you can clarify what this means?

**P7 L25:** Any reason for the continuation? That would even work in f77 in a single line.

**P8 L17:** Technical work has been done to ensure….. As this is a GMD article, I think it would be appropriate to briefly state how you implemented this.

**P10 L7:** Before both routines are called, the tendencies are updated.

This is unclear. What does it mean to update a tendency? Do you mean that the tendencies are applied before these routines are called?

**P11 L16:** Probably should read $c_{O3}$ instead of c.

I am not sure how you decide on the line breaks. I suggest to write the formula in a single line or use a single line for each part/process.

But more importantly you need to define the P and L terms. Otherwise it remains unclear how the derivative of P-L is derived.

**P12 L12:** You need to give units for all variables. Moreover, I would suggest to add the formula for the relative vorticity. This might seem pedantic but it ensures the reproducibility of your work.

**P12 L20:** Multiplication of both. This is ambiguous. It could refer to PV or to the area.

**P12 L20:** Use zonal wind instead of westerly (as you do in L22). Or *absolute zonal* wind if you want to be super precise.

**P12 L32:** middle atmosphere: Please be more precise.

**P13 L10 & L15:** You need to add units to these equations. And I do not get where the 7x is coming from.

**P13 L12:** Please define the mean age of air.

**P13 L22:** … in models and observations. Maybe I am ignorant but what does that have to do with observations?

**P14:** Again give units!

**P14 L15-17:** Please clarify what exactly do you do here? Do you override the default tracer before the microphysics calculation?

**P14 L22:** composition ???

**P15 L11:** 40km does this refer to a triangle side length? Maybe give the grid cell average area for a better comparison to traditional grids.

**P15 L26:** How do the 400DU compare to observations?

**P16 L1:** … losses are positive … This is very difficult to follow. I would suggest: losses are higher in that region.

**P16 L2:** by the chemical mechanism. The plural of chemistry sounds wrong.

**P16 L2:** Passive and reactive tracers. (I think chemical is not the correct word here)

**P16 L4:** And you do not consider additional loss terms (e.g. halogens)

**P16 L6:** negative ozone loss. Is this identical to ozone production? If o it would be better to understand than the double negative formulation.

**P23 Figure 10:** The caption is not correct.

**P23 L10-15:** Please be more precise. Do you mean an absolute decrease or lower values?

e.g. Absolute values of -2E-7 does makes no sense as absolute values are always positive.

**P28 L6:** Age increase of ….. years. It would be nice to give a number here.

**Appendix:** Are all species transported? Or how do you treat e.g. $O^{1D}$. It would be interesting to see the definition for such a short lived species.

6) Language:

**P1 L16:** The age is a measure …

**P4 L6:** (thermos-)dynamics

**P4 L 18:** provide the foundation

**P5 L2:** for XLM file reading.

**P5 L31-33:** That sounds like ozone was hardcoded into two routines. Probably it should read:

In our example, ozone appears in two different….

**P7 L18:** is stored in as

**P8 L21:** reaction schemes

**P8 L32:** shows a schematic

**P10 L9:** computationally

**P11 L9:** ansatz

**P11 L21:** ansatz

**P12 L21:** is given          This sentence needs to be corrected

**P15 L4:** atmospheric composition

**P15 L18:** the TOMS instrument

**P15 L27:** Check this sentence

**P20 L12:** hemispheric

**P21 L11:** zonal is double here

**P22 L2:** Here,  ERA

**P27 L3:** older than in the control

**P27 L31:** different chemical mechanisms

**P27 L31:** extended by additional chemical reactions

**P28 L1:** In the second part

**P28 L6:** Base climatology of ICON

**P28 L7:** climatology of temperature

---

## Author Comment (AC1) · 18 Jul 2018

Dear Dr. Folberth,
Dear referees,

Thank you for handling our submission. Please find our point-by-point response in the supplement. A version of the revised manuscript in a clean version and with the tracked changes will follow.

The main changes and improvements include:

[Figure]

- We included in sections 5.2 and 5.3 a more quantitative analysis.

- Based on the comments by referee 1, we provide a more precise discussion of the differences between the Linoz and the Chapman cycle simulations in section 5.2. Instead of including a scaling factor of 10 to make small changes visible, we decided to plot a zero line separation (gain/loss, also highlighting areas of low temperatures). In addition, we discuss the influence of the relaxation term of LINOZ in more detail.

- We extended the discussion of a timeseries analysis for section 5.3 to emphasise (and quantify) the different aspects of trends and annual variability in the control and feedback simulations of ICON-ART.

- In addition, we compare to TOMS ozone column data as well.

- In section 5.3 onward, we limit our analysis to the period 1990-2009, as suggested by referee 1. The results of the longer simulations can now be found in the appendix

- In general, we revised definitions to be more precise and added missing units.

- We extended the appendix to provide a table of acronyms for convenience.

- Referee 2 suggested that we combine (old) Figure 8&9 as well 10&11. Due to the layout of GMD we decided to stay with separate figures for a clear arrangement.

- The general intention of this paper is to illustrate the status of our ICON-ART development with a strong focus on current technical improvements. We believe,

like referee 2, that the level of technical detail is appropriate for a GMD publication and that the technical aspect is well supported by our use cases.

We hope that we could address the referees concerns adequately and are looking forward to the finalisation of the review process. With kind regards, also on the behalf of all co-authors,

Jennifer Schröter

Please also note the supplement to this comment:
https://www.geosci-model-dev-discuss.net/gmd-2017-286/gmd-2017-286-AC1-supplement.pdf

**Supplement:**

Dear Dr. Folberth,

Thank you for handling our submission. Please find our point-by-point response to the referees comments below. We have included the revised manuscript twice: In a clean version and with tracked changes.

The main changes and improvements include:

- We included in sections 5.2 and 5.3 a more quantitative analysis.

- Based on the comments by referee 1, we provide a more precise discussion of the differences between the Linoz and the Chapman cycle simulations in section 5.2. Instead of including a scaling factor of 10 to make small changes visible, we decided to plot a zero line separation (gain/loss, also highlighting areas of low temperatures). In addition, we discuss the influence of the relaxation term of LINOZ in more detail.

- We extended the discussion of a timeseries analysis for section 5.3 to emphasise (and quantify) the different aspects of trends and annual variability in the control and feedback simulations of ICON-ART.

- In addition, we compare to TOMS ozone column data as well.

- In section 5.3 onward, we limit our analysis to the period 1990-2009, as suggested by referee 1. The results of the longer simulations can now be found in the appendix

- In general, we revised definitions to be more precise and added missing units.

- We extended the appendix to provide a table of acronyms for convenience.

- Referee 2 suggested that we combine (old) Figure 8&9 as well 10&11. Due to the layout of GMD we decided to stay with separate figures for a clear arrangement.

- The general intention of this paper is to illustrate the status of our ICON-ART development with a strong focus on current technical improvements. We believe, like referee 2, that the level of technical detail is appropriate for a GMD publication and that the technical aspect is well supported by our use cases.

We hope that we could address the referees concerns adequately and are looking forward to the finalisation of the review process. With kind regards, also on the behalf of all co-authors,

Jennifer Schröter

Dear referee, Thank you for your review of the paper. Your comments and remarks helped us to clarify and improve the manuscript. Your feedback provided us with the opportunity to extend certain aspects of our discussion - much appreciated! Our basic idea in writing this paper was to give a status-quo of our model development. We decided that GMD is the right place to give a detailed overview of our implementation. Most user guides, as you suggest, are somehow non citable literature. Our approach of combining a structure like XML with a Fortran interface should be seen as a blueprint for other modelling frameworks as well and supports us today in the flexible setup of model integrations for science applications.

In the following, you can find our detailed answers (black) to your comments (red).

Page 2, Lines 31-32: I might suggest changing the text from field, like large eddy simulations, numerical weather predictions and climate simulations ICON-ART can run with the different existing physics configurations. to field, like large eddy simulations, numerical weather predictions or climate simulations, ICON-ART can run with different existing physics configurations.

Thank you for this suggestion. We rephrased this sentence.

Page 4 Lines 6-7: Should the phrase '...not have any impact on other tracers of the (thermo-)dynamical of the simulated system.' be '... not have any impact on other tracers or the (thermo-)dynamics of the simulated system.'

Yes, thank you. This sentence got a little bit mixed up.

Page 13, Lines 5-16: Age of air derived from a linearly increasing tracer is fairly widely used and I do not think all the details are required of the implementation. But I would also note that equation (4) is a bit difficult to understand. Is the first term on the right-hand side (7 x 86400.0 x 365.2425) the initial value assigned to the age tracer every- where in the model?

We made a mistake in typesetting this formula (4 - now corrected). The age of air tracer is initialised with a value of 7 years (in seconds) in the troposphere. This value is scaled with the respective pressure. For all other time steps, the term on the right hand side (7 x 86400 x 365) is the lower boundary condition for altitudes below 950 hPa and each integration time step the integration time step is added.

Can the authors provide some reasons why total column ozone inside the vortex is lower while chemical change at 50 hPa suggests it should be higher? I am also curious about the ozone loss that occurs outside the vortex in the Linoz simulation and that is mentioned in this section as well. Is it possible to add the 195K isotherm on to Figure 4 to help the reader unambiguously see where the PSC-chemistry parameterisation is active? While I

can imagine some filaments of the original vortex having temperatures below 195K, I am curious why the two lobes of the original vortex are not cold enough to activate the ozone loss parameterisation.

We have now shaded the area below 195 K in Figure 4. In addition, we added additional information on the influence of the LINOZ background ozone climatology. It should be noted that the background climatology shows higher values of ozone volume mixing ratios at 50 hPa in comparison to the ERA-Interim initialisation that we used. In the southern hemisphere, at 50 hPa, this differences is up to $2e-6\text{mol mol}^{-1}$. The relaxation towards the background climatology produces a somehow pseudo production term. Referring to the original paper McLinden et al. (2000), the photochemical relaxation time would be up to 100 days. Since we are performing a short term proof-of-concept simulation, the relaxation to the (higher) background climatology manifests itself in a pronounced production and loss pattern. For future forecasts we would envisage an initialisation phase in which background and initial ozone are calibrated before the forecast, leading to a smaller amplitude. However, additional loss is added by the parameterisation of heterogeneous processes based on low temperatures, but it is less visible. For climate simulations (below) this is not a problem, because an initial spin-up phase is included.

For clarity, we decided to plot active and passive ozone separately as well (Figure X). For the difference plot, we neglected the scaling factor of 10, for the Chapman cycle (highlighting that the tendencies are generally small). In the original vortex lobes, temperatures below 195 K do occur, but only for a very short time.

Thank you very much for pointing out ambiguities. We hope our clarifications (based on your suggestions) will help the reader to understand the workings of the model better.

.

Page 17, Lines 3-4: For the control simulation that uses specified ozone from Cionni et al. (2011) in the radiation, what water vapour is used for radiation? Is it the water vapour that crosses the tropopause, but with no addition from methane oxidation?

The water vapour is the standard water vapour tracer of ICON. This water vapour tracer is transported and is part of the microphysics scheme. There is no vertical limit to which this tracer is present, thus, this water vapour tracer is also transported across the tropopause. If the cold point is reached, this water vapour tracer undergoes the transition to the ice tracer.

Page 19, Lines 9 14: The standard deviation of ozone is discussed here and the point that the variability in the Cionni et al. dataset shows a pattern very similar to the ozone contours themselves is made. The Cionni et al. ozone dataset was made using a standard climatological ozone with a temporal trend given by regression to an idealised reactive chlorine (Cly). I think what you are showing in the standard deviation is just the long-term trend in ozone itself the evolution of the trend around the long-term mean for the 1980 2009 period. While the Linoz ozone actually has year-to-year variability but as it is constructed I dont think it has any long-term trend. This comparison does not seem very valid. And that brings up another point that since Linoz is designed for a particular time period, particularly the PSC parameterization, is it correct to use it over the full

period from 1980 when chlorine loading will have evolved considerably? I might suggest restricting the comparison to 1990-2009 when the effects of ODSs are more fully realized.

Thank you very much for suggesting a clarification. Your are completely right, pointing out that the ozone climatology used in ICON-ART control has a long term trend. In contrast, the ICON-ART feedback simulation with Linoz does not include a trend term in the ozone (it could be included by time varying coefficients). To clarify this and make results even more comparable, we took up your suggestion to concentrate the discussion on the years 1990-2009 and changed the analysis period accordingly. To provide a more quantitative discussion, we added a table with the results of a timeseries analysis of the total ozone column at 60°S from the control and feedback integrations and compare both to TOMS observations.

Page 20, Line 5: In the southern hemisphere winter, ICON-ART reaches temperatures of about 200 K. This seems to be like a significant warm bias that would have serious impacts on PSC processes. How does this statement agree with the findings shown in Figure 9 that the feedback simulation seems to be biased cold compared to ERA- Interim?

The line you are referring to has been changed, because it was too vague. You are right that the feedback simulation is cold biased in the southern hemisphere. The control simulation shows this as well but lies in between ERA-Interim and the feedback simulation.

Page 20, Lines 7-8: Is the comparison reversed in the statement In the vertical region around 50 hPa, the difference between ERA-Interim and the feedback simulation is about 15 K to 20 K in the tropics and below 8 K in the southern hemisphere.

Thank you for pointing this out - the statement has been corrected.

Page 22, Lines 2-3: On the comparison of zonal winds to ERA-Interim the authors state Here, the ERA-Interim shows values up to 25 m s1 higher than in the ICON- ART simulation. Between 30N and 60N latitude above 20 hPa the sign of the differences changes. Here, we observe stronger zonal wind speeds than in ERA- Interim. The finding of stronger zonal winds, particularly around the southern hemisphere stratospheric jet, does not seem to fit with the cold temperatures found the ICON-ART feedback simulation that are shown in Figure 9. I think the negative difference means the winds in ICON-ART are stronger.

We corrected and clarified the statement - thank you!

Page 23  the caption on Figure 10 does not seem to match with what is shown in Figure 10.

Corrected - thank you!

 It is difficult to see the difference in water vapour entering the stratosphere that is mentioned by Due to lower tropical tropopause temperatures in the feedback simulation, less water can enter the lower stratosphere. because Figure 12 is the monthly anomaly to the annual mean. A quick mention of the difference in the annual mean in the lower stratosphere would help.

Now, we mentioned the annual mean temperature difference between the ICON-ART control and feedback simulations in the text.

  Do you have any idea why the maximum in the annual cycle in the feedback simulation around 80 hPa is so broad, stretching from May to October. The broad annual cycle in the bottom right corner of the feedback plot is also difficult to reconcile with the very narrow maximum in the top left corner.

The tropopause temperature differences do correspond to that. From JJA to SON, the temperature in the tropics around 100 hPa are higher in the feedback simulation than in the control simulation. We extended the respective section in the paper by discussing this aspect.

Dear Johannes Bieser, Thank you for your review of our paper. We extended section 5.2 providing additional information, focusing on details of the Linoz parameterisation.

For section 5.3, we included a detailed timeseries analysis to make the comparison more quantitative, keeping your remark in mind.

All abbreviations can now be found in the appendix. We added missing units and clarified definitions. We certainly agree, that this improves reproducibility.

We have considered your remarks regarding combining figures and to alter line breaks. However, on balance we found that given the final publication style of GMD the current layout should work well. If problems are encountered in the layout process, we will happily adapt the layout.

In the following, you can find our detailed answers (black) to your comments (red).

P2 L5: ... the same dynamical core .... As you know (and state later in the text) this is not the case for ICON (and I am not sure which other model has actually reached that ideal).

We agree that ICON is not a uniform model in terms of using the same physical parameterisations across scales. However, ICON uses one dynamical core across scales (like, e.g. the UM at the Met Office does as well).

P2 L22: Here you should mention that the development is based on COSMO-ART. Maybe I am wrong but people do know COSMO-ART. In this case add a few sentences to clarify any differences between the ART in COSMO-ART and ICON-ART

We included an additional sentence. Indeed, the main development goals are shared. Nevertheless, parameterisations and code implementations are improved or very different.

P7 L6: Here I got lost: prognostic and diagnostic state. Maybe you can clarify what this means?

We removed that sentence and replaced it by: The ART tracers are part of the general tracer transport scheme in ICON.

P7 L25: Any reason for the continuation? That would even work in f77 in a single line. Yes, you are completely right. The final version of this paper will be published in a two-column layout. This line would be too long for that. The continuation is only a beautification for GMD.

P8 L17: Technical work has been done to ensure..... As this is a GMD article, I think it would be appropriate to briefly state how you implemented this.

We extended this section by two additional sentences.

P10 L7: Before both routines are called, the tendencies are updated. This is unclear. What does it mean to update a tendency? Do you mean that the tendencies are applied before these routines are called?

We rephrased that sentence. Instead of returning tendencies, the routines apply the calculated tendencies and change or alter the respective (state) variable, e.g. the water vapour tracer.

P11 L16: Probably should read cO3 instead of c. I am not sure how you decide on the line breaks. I suggest to write the formula in a single line or use a single line for each part/process. But more importantly you need to define the P and L terms. Otherwise it remains unclear how the derivative of P-L is derived.

We defined these terms more precisely. Here, again, we decided to stay with the line breaks with respect to GMD layout.

P12 L12: You need to give units for all variables. Moreover, I would suggest to add the formula for the relative vorticity. This might seem pedantic but it ensures the reproducibility of your work.
We extended the respective section.

P12 L20: Multiplication of both. This is ambiguous. It could refer to PV or to the area. Yes, you are right. This sentence is a little bit too complicated. To be more precise, we separated this into two sentences.

P12 L20: Use zonal wind instead of westerly (as you do in L22). Or absolute zonal wind if you want to be super precise.

Thank you - corrected!

P12 L32: middle atmosphere: Please be more precise.

We added an altitude constraint.

P13 L10 & L15: You need to add units to these equations. And I do not get where the 7x is coming from.

We improved the discussion of this formula.

P13 L12: Please define the mean age of air.

We extended the definition by defining the unit of $\Psi_{age}$. This also allows to clarify the origin of the given number of 7 years in the equation (given in seconds).

P13 L22: ... in models and observations. Maybe I am ignorant but what does that have to do with observations?

We rephrased this section.

P14: Again give units!
We extended our definitions by providing the units of all variables.

P14 L15-17: Please clarify what exactly do you do here? Do you override the default tracer before the microphysics calculation?

We clarified that sentence.

P14 L22: composition ???
We added a missing *chemical*.

P15 L11: 40km does this refer to a triangle side length? Maybe give the grid cell average area for a better comparison to traditional grids.

The length of 40 km is a nominal characteristic distance computed as square-root of the mean cell area of the grid. A reference for a formula has been added.

P15 L26: How do the 400DU compare to observations?

We extended this section by a more detailed comparison to the observations.

P16 L1: ... losses are positive ... This is very difficult to follow. I would suggest: losses are higher in that region.
We totally agree and changed that.

P16 L2: by the chemical mechanism. The plural of chemistry sounds wrong.
We corrected that.

P16 L2: Passive and reactive tracers. (I think chemical is not the correct word here)

We agree.

P16 L4: And you do not consider additional loss terms (e.g. halogens)
We added a clarifying sentence.

P16 L6: negative ozone loss. Is this identical to ozone production? If o it would be better to understand than the double negative formulation.
You are right, we clarified that.

P23 Figure 10: The caption is not correct.
Thank you for pointing this out. We corrected the mistake.

P23 L10-15: Please be more precise. Do you mean an absolute decrease or lower values? e.g. Absolute values of -2E-7 does makes no sense as absolute values are always positive.

Yes, we improved the phrasing.

P28 L6: Age increase of ..... years. It would be nice to give a number here.
We extended the description.

Appendix: Are all species transported? Or how do you treat e.g. O1D. It would be interesting to see the definition for such a short lived species.

Species with a short lifetime are not transported. Information about that is stored in a look-up table of MECCA. We extended that example.

---

## Author Response (AR2)

Dear Dr. Folberth,
Thank you for your effort in handling our manuscript. We had a look at the suggested changes.

Page 2, Line 2  missing word in climate models [were] built upon...

We corrected that mistake - thank you!

Page 21, Table 3  The values for the feedback run over 1980-1997 is 1.314  0.036 DU yr-1, but the trend is in red as being not statistically significant. The trend is   15 times larger than the standard error, so isnt this trend likely to be statistically significant? I might be wrong but I just wanted to note it as worthy of checking.

We apologise for the confusion. We used a far too complicated trend model. We simplified our approach and updated Table 3. We now only provide the trend values and the RMSE and do not discuss explicitly significance for the short time periods. The simplified trend model now only includes the removal of the annual cycle and a linear fit to the residuum.

Page 27, Figure 13  Do you have any explanation for why the magnitude of the annual cycle in water vapour seems to become larger at higher levels? The ERA-Interim plot shows a maximum amplitude in the lower stratosphere with a decreasing amplitude with height and this is the expected pattern as mixing of air into the tropical pipe reduces the amplitude. But the ICON-ART simulations, and particularly the feedback simulation, shows the annual cycle being a maximum at the top.

We have revised the text and included a explanatory link to the methane oxidation scheme. With kind regards, also on the behalf of all co-authors,

Jennifer Schröter

[revised manuscript text omitted]